# Cortical anchoring of the microtubule cytoskeleton is essential for neuron polarity

**Liu He[1†], Robbelien Kooistra[1†], Ravi Das[2†], Ellen Oudejans[1†], Eric van Leen[1], Johannes Ziegler[3], Sybren Portegies[1], Bart de Haan[1], Anna van Regteren Altena[1], Riccardo Stucchi[1,4], AF Maarten Altelaar[4], Stefan Wieser[3], Michael Krieg[2], Casper C Hoogenraad[1,5]\*, Martin Harterink[1]\***

[1]Cell Biology, Neurobiology and Biophysics, Department of Biology, Faculty of Science, Utrecht University, Utrecht, Netherlands; [2]Neurophotonics and Mechanical Systems Biology, ICFO-Institut de Ciencies Fotoniques, The Barcelona Institute of Science and Technology, Barcelona, Spain; [3]Fast live-cell superresolution microscopy, ICFO-Institut de Ciencies Fotoniques, The Barcelona Institute of Science and Technology, Barcelona, Spain; [4]Biomolecular Mass Spectrometry and Proteomics, Bijvoet Center for Biomolecular Research and Utrecht Institute for Pharmaceutical Sciences, Utrecht University, Utrecht, Netherlands; [5]Department of Neuroscience, Genentech, Inc, South San Francisco, United States

**\*For correspondence:**
c.hoogenraad@uu.nl (CCH);
m.harterink@uu.nl (MH)

[†]These authors contributed equally to this work

**Competing interests:** The authors declare that no competing interests exist.

**Abstract** The development of a polarized neuron relies on the selective transport of proteins to axons and dendrites. Although it is well known that the microtubule cytoskeleton has a central role in establishing neuronal polarity, how its specific organization is established and maintained is poorly understood. Using the in vivo model system *Caenorhabditis elegans*, we found that the highly conserved UNC-119 protein provides a link between the membrane-associated Ankyrin (UNC-44) and the microtubule-associated CRMP (UNC-33). Together they form a periodic membrane-associated complex that anchors axonal and dendritic microtubule bundles to the cortex. This anchoring is critical to maintain microtubule organization by opposing kinesin-1 powered microtubule sliding. Disturbing this molecular complex alters neuronal polarity and causes strong developmental defects of the nervous system leading to severely paralyzed animals.

## Introduction

The structural organization and dynamic remodeling of the cytoskeleton is essential for neuronal development and survival. Axons and dendrites are filled with bundles of non-centrosomal microtubules, which are important for maintaining cellular structure as well as intracellular transport. Notably, the microtubule cytoskeleton plays a central role in neuronal development and polarization: axons have microtubules with their plus-end projecting away from the soma, while dendrites have either a mixed (vertebrates), or uniform minus-end out (invertebrates), microtubule organization (*Baas et al., 1989*; *Kapitein and Hoogenraad, 2015*). This difference in microtubule organization allows for selective cargo transport to either neurite by specific microtubule motors and is considered an evolutionary conserved characteristic of axon-dendrite polarity (*Rolls and Jegla, 2015*). Disorganized axonal and dendritic microtubules lead to deficient transport and subsequently to neuronal polarity defects (*Harterink et al., 2018*; *Yogev et al., 2016*). In addition, defects of the microtubule cytoskeleton are associated with a range of neuronal diseases (*Falnikar and Baas, 2009*; *Franker and Hoogenraad, 2013*) and the capacity of neurons to regenerate is intimately

linked to microtubule stability and plus-tip dynamics (*Blanquie and Bradke, 2018*; *Tang and Chisholm, 2016*).

During early neuronal development, microtubules are relatively mobile and can be transported by microtubule motors. Studies in *Drosophila* and *Caenorhabditis elegans* indicate that the motor kinesin-1 has a role in setting up the characteristic axon-dendrite microtubule organization, by sliding microtubules against each other (*Lu et al., 2013*; *Winding et al., 2016*; *Yan et al., 2013*). At later stages of development, the entire microtubule cytoskeleton is largely immobilized (*Kahn et al., 2018*; *Lu et al., 2013*), which is important to maintain the specific microtubule organization. Various microtubule associated proteins (MAPs) bind to the microtubule lattice, which can stabilize the microtubule itself and can cross-link them together thus preventing microtubule-microtubule sliding (*Bodakuntla et al., 2019*). In addition, the microtubule bundles were found connected to other cortical cytoskeletal elements (*Fréal et al., 2016*; *Liang et al., 2013*; *Qu et al., 2017*). Although the importance of cortical anchoring of microtubule bundles to maintain neuronal microtubule organization and thereby maintain neuronal polarity and function is apparent, the evidence for this is largely lacking.

In this study, we use *C. elegans* to identify the molecular mechanisms underlying neuronal microtubule organization. Surprisingly, we found that the highly conserved protein UNC-119 forms a ternary complex with UNC-44 (Ankyrin) and UNC-33 (CRMP) in vitro and in vivo. We show that this complex forms a similar periodic arrangement as the spectrin cytoskeleton and that it is critical to anchor the microtubule cytoskeleton to the neuronal cortex. In the absence of cortical anchoring, UNC-116 (kinesin-1) induces massive microtubule cytoskeleton sliding in axons and dendrites, leading to loss of axon-dendrite microtubule polarity. Thus, we propose that the balance between kinesin-1 dependent microtubule sliding and cortical microtubule anchoring is essential for axon-dendrite microtubule polarity and thus for neuron development and functioning.

## *unc-119* is important for neuronal polarity and development

Genetic screens in *C. elegans* identified many *unc*-genes, which lead to impaired ('*unc*coordinated') locomotion (*Brenner, 1974*). Neuronal polarity defects and disorganized neuronal microtubules have been observed in several *unc*-mutants (*Maniar et al., 2012*; *Yan et al., 2013*). In this study, we analyzed neuronal development and axon-dendrite polarity in the highly polarized PVD neuron (*Figure 1A*), which is widely used as a model neuron (*Sundararajan et al., 2019*). We found that the *unc-119* mutant *ed3*, which carries an early stop mutation, has a fully penetrant microtubule orientation defect in the PVD anterior dendrite (*Figure 1B and C*). Axons were occasionally affected as well (*Figure 1—figure supplement 1A*). The microtubule orientation defect is not specific to the PVD neuron, as we observed a similar phenotype in the PHC neuron (*Figure 1—figure supplement 1B*). In contrast, the ciliated URX and PQR neurons did not have obvious microtubule defects (*Figure 1—figure supplement 1C and D*), as is expected because these neurons have additional mechanisms to organize neuronal microtubules (*Harterink et al., 2018*). Furthermore, synaptic vesicles were consistently mislocalized to the cell body and dendrites in the *unc-119* mutant (*Figure 1D* and *Figure 1—figure supplement 1E–I*), and we observed morphological defects for the PVD neuron; the axon and anterior dendrite are shorter and dendrite branching is reduced (*Figure 1E* and *Figure 1—figure supplement 1J–L*). The disorganized microtubules and vesicle mislocalization resemble defects previously observed in *unc-33* (CRMP) and *unc-44* (Ankyrin) mutants (*Maniar et al., 2012*). *unc-119* is mainly expressed in neurons (*Maduro and Pilgrim, 1995*) and codes for a highly conserved protein (*Maduro et al., 2000*), homologous to human UNC119A and UNC119B. Indeed, by introducing UNC-119::GFP specifically in the PVD neuron we were able to suppress the dendritic microtubule defect (*Figure 1C*), which indicates that UNC-119 functions in a cell intrinsic manner. However, the molecular function of UNC-119 in controlling neuronal development is completely unknown.

To study UNC-119 protein localization, we generated a C-terminal GFP knock-in line using the SEC-approach (*Dickinson et al., 2015*) and observed GFP expression in all neurons (*Figure 1F*), as previously reported (*Lim et al., 1999*; *Maduro and Pilgrim, 1995*). The GFP signal can be detected in axons (ventral nerve cord and nerve ring) as well as dendrites (amphid nerve). Strikingly, the protein is not enriched in the cell bodies, which suggests that the protein is not freely diffusing in the cytoplasm. To determine UNC-119 protein dynamics, we performed fluorescence recovery after photobleaching (FRAP) microscopy (*Diegelman, 1998*). We found that both in axons and dendrites the fluorescence hardly recovers and that the immobile UNC-119 pool is 70–80% (*Figure 1G–I*).

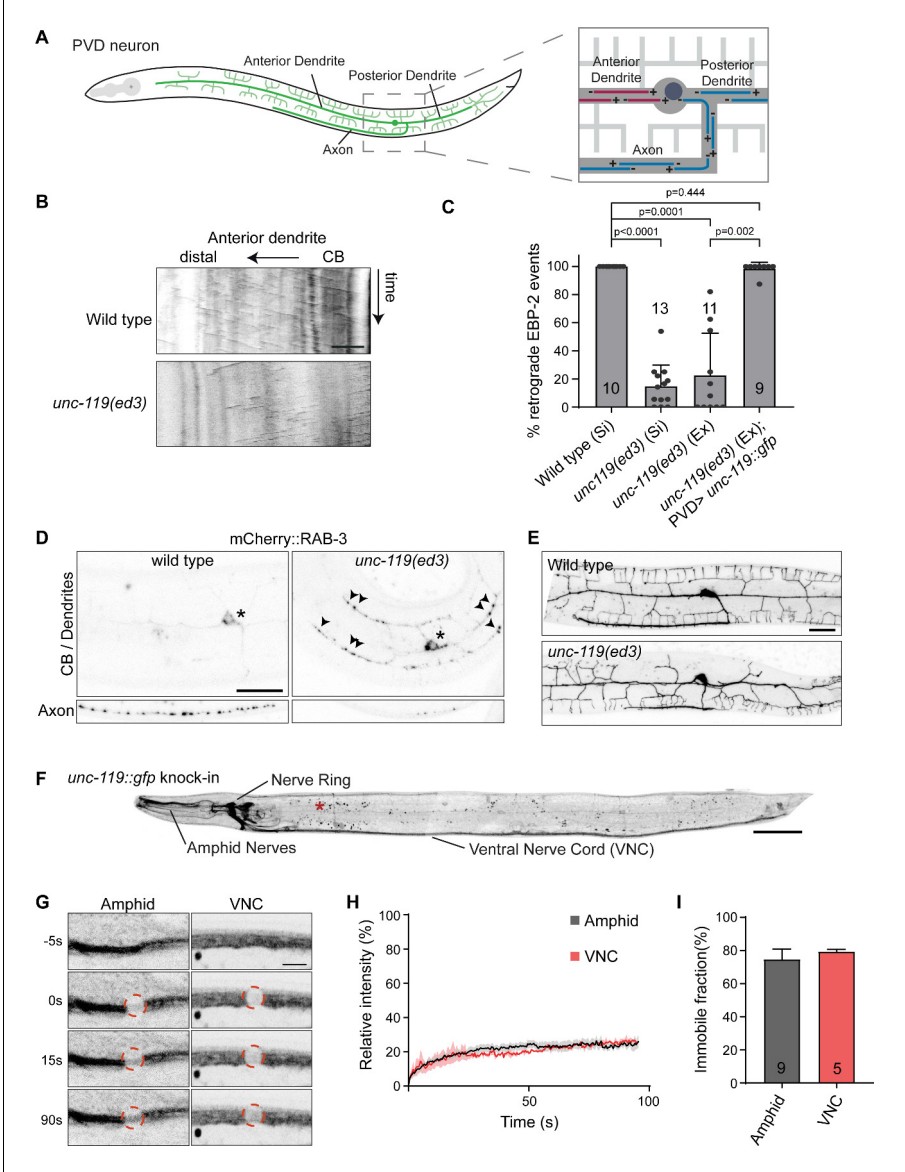

**Figure 1.** *unc-119* is essential for neuronal microtubule organization. (**A**) Schematic representation of the PVD neuron and its microtubule organization; it possesses two highly branched sensory dendrites and a single axon which connects to other neurons in the ventral nerve chord. Only the anterior dendrite has minus-end out microtubules (red), whereas the axon and posterior dendrites have plus-end out microtubules (blue) (*Harterink et al., 2018*). (**B–C**) Representative kymographs and quantification of the microtubule orientation in the PVD anterior dendrite. EBP-2::GFP was expressed in the PVD to visualize growing microtubules using the *Pdes-2* promotor. EBP-2::GFP strains were single copy integration lines (Si) generated using mosSCI or were extrachromosomal lines (Ex). For *unc-119 (Si)* we used CRISPR to mutate the *cb-unc-119* from the mosSCI generated line (wild type (Si)). Number of analyzed animals is indicated. Scale, 5 µm. (**D**) Representative images of synaptic vesicle localization in the PVD neuron marked by RAB-3::mCherry. Cell body (CB) and dendrites are separately shown from the axon to minimize the autofluorescence. Asterisk, marks the cell body; arrowheads point to RAB-3::mCherry in the dendrites; scale, 20 µm. (**E**) Representative image of the PVD morphology in wild type and the *unc-119* mutant. Scale, 20 µm. (**F**) *unc-119::gfp* knock-in animals. Expression is strongest in neurons, but can also be detected at low levels in non-neuronal cells such as the vulva cells and the seam cells. The locomotion of the *gfp* knock-in animals is not affected (*Figure 1—figure supplement 1M*), indicating that the fusion protein is functional. The red asterisk marks autofluorescent gut granules (small puncta) seen throughout the animal. Image were straightened using in imageJ. Scale, 50 µm. (**G**) UNC-119::GFP FRAP in a dendrite bundle (Amphid) and axon bundle (VNC). Scale, 5 µm. (**H–I**) Average normalized intensity graph of UNC-119::GFP FRAP ± SEM (**H**) and the

*Figure 1 continued on next page*

*Figure 1 continued*

percentage immobile fractions after FRAP (I). Number of analyzed animals is indicated. Analyzed animals were from the L4 or young adult stage; error bars represent SD; statistical analysis, Kruskal-Wallis test followed by Dunn's multiple comparisons test.

The online version of this article includes the following source data and figure supplement(s) for figure 1:

**Source data 1.** Source data for graphs in *Figure 1*.
**Figure supplement 1.** Quantifications of the *unc-119* mutant phenotypes.
**Figure supplement 1—source data 1.** Source data for graphs in *Figure 1—figure supplement 1*.

Taken together, this suggests that UNC-119 is tightly associated with undefined structures in axons and dendrites, where it is important for neuronal development and microtubule organization.

## UNC-119 forms a complex with UNC-44 (Ankyrin) and UNC-33 (CRMP)

To understand how UNC-119 functions in organizing neuronal polarity, we performed immunoprecipitations experiments from *C. elegans* using the *unc-119::gfp* knock-in animals followed by mass spectrometry to identify potential interaction partners of UNC-119 (*Supplementary file 1*). Interestingly, one of the top-scoring partners of UNC-119, UNC-33 (CRMP), was reported to be important for neuronal development and microtubule organization (*Maniar et al., 2012*) and mutations in *unc-33* lead to similar microtubule defects (*Harterink et al., 2018*) as the *unc-119* mutant (*Figure 1* and *Figure 1—figure supplement 1*). UNC-33 is a putative microtubule binding protein, which has three isoforms in *C. elegans*: small (UNC-33S), medium (UNC-33M) and large (UNC-33L), but only the large isoform is able to rescue the neuronal polarity defects of the *unc-33* mutant (*Maniar et al., 2012*; *Tsuboi et al., 2005*). To validate and map the interaction between UNC-33 and UNC-119, we performed pull-down experiments in cultured Hek293T cells and found that the UNC-33L, but not UNC33S, efficiently binds to UNC-119 (*Figure 2A and B*). Moreover, UNC-119 was able to pull down the UNC-33L specific N-terminus (*Figure 2B*), further proving that the UNC-119 interacts with the UNC-33L isoform.

Previous work has shown that UNC-33 (CRMP) acts downstream of *unc-44* (Ankyrin) and that the *unc-44* mutant has very similar phenotypes as the *unc-33* mutant (*Harterink et al., 2018*; *Maniar et al., 2012*; *Meng et al., 2016*). However, we failed to detect a direct interaction between UNC-44 and UNC-33 by immunoprecipitation from Hek293T cells (*Figure 2D*). In a yeast two-hybrid screen for *C. elegans* interacting proteins, UNC-119 was identified as a possible binding partner for UNC-44 (*Li et al., 2004*). This, together with our finding that UNC-119 interacts with UNC-33, suggests that UNC-119 bridges UNC-44 to UNC-33. To test the possible interaction between UNC-44 and UNC-119, we cloned UNC-44 into six fragments (*Figure 2A*). Using these fragments in Hek293T cells we found that the UNC-44 C-terminus specifically was able to bind to UNC-119 (*Figure 2C*), which is in agreement with the interaction domain identified by the yeast two-hybrid screen (*Li et al., 2004*). To test if UNC-44/UNC-119/UNC-33 form a ternary complex we performed pull-down experiments with the UNC-44 C-terminus mixed with either UNC-119::GFP, UNC-33L::GFP or both proteins. We again found that UNC-119 efficiently binds to the UNC-44 C-terminus, however UNC-33L only binds to UNC-44 in the presence of UNC-119 (*Figure 2D*), showing that UNC-33L, UNC-44, and UNC-119 together can form a ternary complex (*Figure 2E*). Since mutants for each of these three *unc*-genes have very similar phenotypes and that the corresponding proteins can form a complex strongly suggests that UNC-33L, UNC-44, and UNC-119 have a common function in neuronal development.

## UNC-44 (Ankyrin) immobilizes UNC-33 (CRMP) and UNC-119 in neurons

To determine whether the ternary complex also exists in living organisms, we visualized the localization and dynamics of the individual proteins. Using CRISPR/Cas9, we inserted a GFP at the C-terminus of the giant AO13 Ankyrin splice isoform, which is essential for neuronal development (*Boontrakulpoontawee and Otsuka, 2002*; *Otsuka et al., 2002*). UNC-44::GFP mainly localizes to axons and dendrites and its expression pattern is nearly indistinguishable from the UNC-119::GFP knock-in (*Figure 3B*). We observed that UNC-44::GFP clearly localizes to the cortex (*Figure 3C*), and that it is highly immobile by FRAP in axons and dendrites (*Figure 3D–F*), as expected from a protein

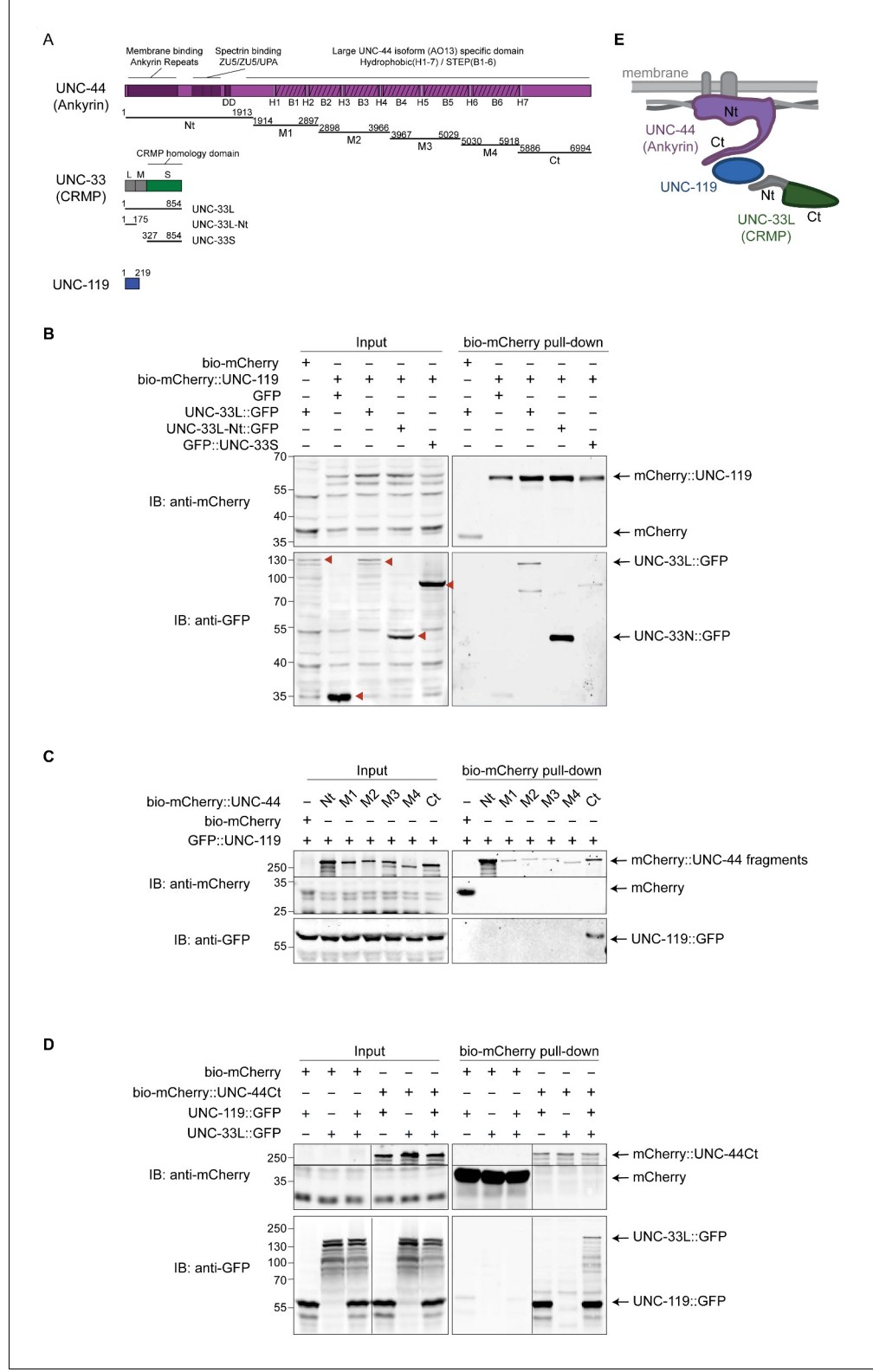

**Figure 2.** UNC-119 forms a complex with UNC-44 (Ankyrin) and UNC-33L (CRMP). (**A**) Schematic representation of the protein and fragments used for pull-down experiments. *C. elegans* expresses a giant UNC-44 (Ankyrin) isoform (AO13) that is mainly expressed in neurons (*Otsuka et al., 2002*) and which contains six blocks of serine/threonine/glutamic acid/proline rich (STEP) repeats separated by seven hydrophobic domains (H) in its C-terminus. Three isoforms exist for UNC-33 (CRMP). The N-terminal extensions of UNC-33M and UNC-33L are marked in

*Figure 2 continued on next page*

*Figure 2 continued*

grey. Nt = N terminus; Ct = C terminus; DD = death domain. (**B–D**) Streptavidin pull-down assays were performed by incubating streptavidin coated magnetic beads with lysates of HEK293T cells (co)expressing the indicated fusion-constructs for 1 hr at 4°. (**B**) Pull-down from lysates of cells coexpressing BirA with either bio-mCherry or bio-mCherry-UNC-119 (bait) and the indicated GFP labeled proteins (prey). The GFP labeled proteins are indicated by red arrowheads in the input blot. (**C**) Pull-down of bio-mCherry labeled UNC-44 fragments and UNC-119::GFP. Proteins were separately expressed in HEK293T cells (UNC-44 fragments required longer incubation time for expression). The lysates were mixed before the pull-down. We noticed that the UNC-44 fragments (M1–M4 and Ct) run at higher Mw than expected, which could be caused by post translational modifications. (**D**) Pull-down of bio-mCherry or bio-mCherry labeled UNC-44 C-terminus (Ct) with UNC-119::GFP and/or UNC-33L::GFP. mCherry proteins were separately expressed from the GFP proteins. Lysates were mixed before the pull-down. (**E**) Interaction model based on the pull-down experiments; where UNC-44 (Ankyrin) is drawn associated to membrane proteins (top) and spectrin (side).

that is part of cortical cytoskeleton (*Bennett and Lorenzo, 2016*), Moreover, this immobility does not depend on either UNC-119 or UNC-33 (CRMP) (*Figure 3G and H*, *Figure 3—figure supplement 1C and D*), which is expected if UNC-44 (Ankyrin) acts upstream of these proteins. However, when we analyzed the dynamics of the UNC-119::GFP in the *unc-44* mutant, we found that the UNC-119 recovers rapidly and seems to diffuse freely in the absence of UNC-44 (*Figure 3I and J*, *Figure 3—figure supplement 1E and F*). Notably, the UNC-119 immobility was not affected in the *unc-33* mutant. Taken together, this data suggests that UNC-44 binds and stabilizes UNC-119.

Since UNC-33 exists as three isoforms and that an N- or C-terminal GFP fusion of UNC-33L disrupts protein function (results not shown), it is not possible to endogenously tag specific isoforms. However, inserting an internal GFP resulted in a functional UNC-33L protein (*Figure 3K* and *Figure 3—figure supplement 1G*; *Maniar et al., 2012*). Therefore, we generated single copy (MosSCI) transgenes for each of the UNC-33 isoforms, which we expressed in the PVD neuron, and found that specifically UNC-33L is able to rescue the neuronal microtubule orientation defect of the *unc-33* mutant (*Figure 3M*). We did not detect a specific subcellular localization for UNC-33L, nor did we observe obvious differences in localization between the three UNC-33 isoforms, except for UNC-33S and UNC-33M being a bit enriched in the cell body (*Figure 3L*; *Maniar et al., 2012*). However, FRAP analysis did reveal differences in dynamic properties between the UNC-33 isoforms. UNC-33L signal did hardly recover (immobile fraction is ~85%; *Figure 3N–P*), whereas UNC-33M and UNC-33S partially recovered (immobile fraction is ~60%). This suggests that the two smaller UNC-33 isoforms have a rapidly diffusing pool as well as an anchored pool. Furthermore, we found that the UNC-33L immobility is completely lost in the *unc-44* or *unc-119* mutants (*Figure 3Q and R*; *Figure 3—video 1*), which indicates that UNC-33 acts downstream of UNC-44 and UNC-119. Similarly, GFP::UNC-33S mobility increases in the *unc-119* and *unc-44* mutant backgrounds (*Figure 3S-T*). In conclusion, these results together with the biochemical interaction data suggest that UNC-44 (Ankyrin) immobilizes UNC-33L (CRMP) via UNC-119.

## UNC-44 (Ankyrin), UNC-119 and UNC-33 (CRMP) form a periodic pattern along neurites

Ankyrin proteins are well-characterized component of the cortical actin-spectrin cytoskeleton (*Bennett and Baines, 2001*), which forms a characteristic periodic cytoskeleton in invertebrate (*Krieg et al., 2017*) and vertebrate neurons (*Xu et al., 2013*). We thus sought to investigate if UNC-44, similar to Ankyrin-G in mammalian and fly neurons (*Leterrier et al., 2015*; *Pielage et al., 2008*) forms periodic structures as an indicator for its integration into the cortical cytoskeleton. In order to super resolve the cytoskeleton, we resorted to TIRF-SIM on isolated primary neuron cultures (Materials and methods, *Figure 4—figure supplement 1*). We observed that UNC-44::GFP (Ankyrin) consistently forms a periodic assembly similar to the 180–200 nm pattern observed with YFP::UNC-70 (β-spectrin) (*Figure 4A*; p=0.17, pairwise Wilcoxon rank sum test). Importantly, we also observed a ~ 200 nm periodic pattern with the UNC-119::GFP, that corresponds well with the observed UNC-44 distribution (*Figure 4B*; p=0.82, pairwise Wilcoxon rank sum test). Next, we isolated primary neurons from transgenic animals for the full-length, long version of UNC-33L, which we found to interact with UNC-119, and as well the truncated UNC-33M version, which does not

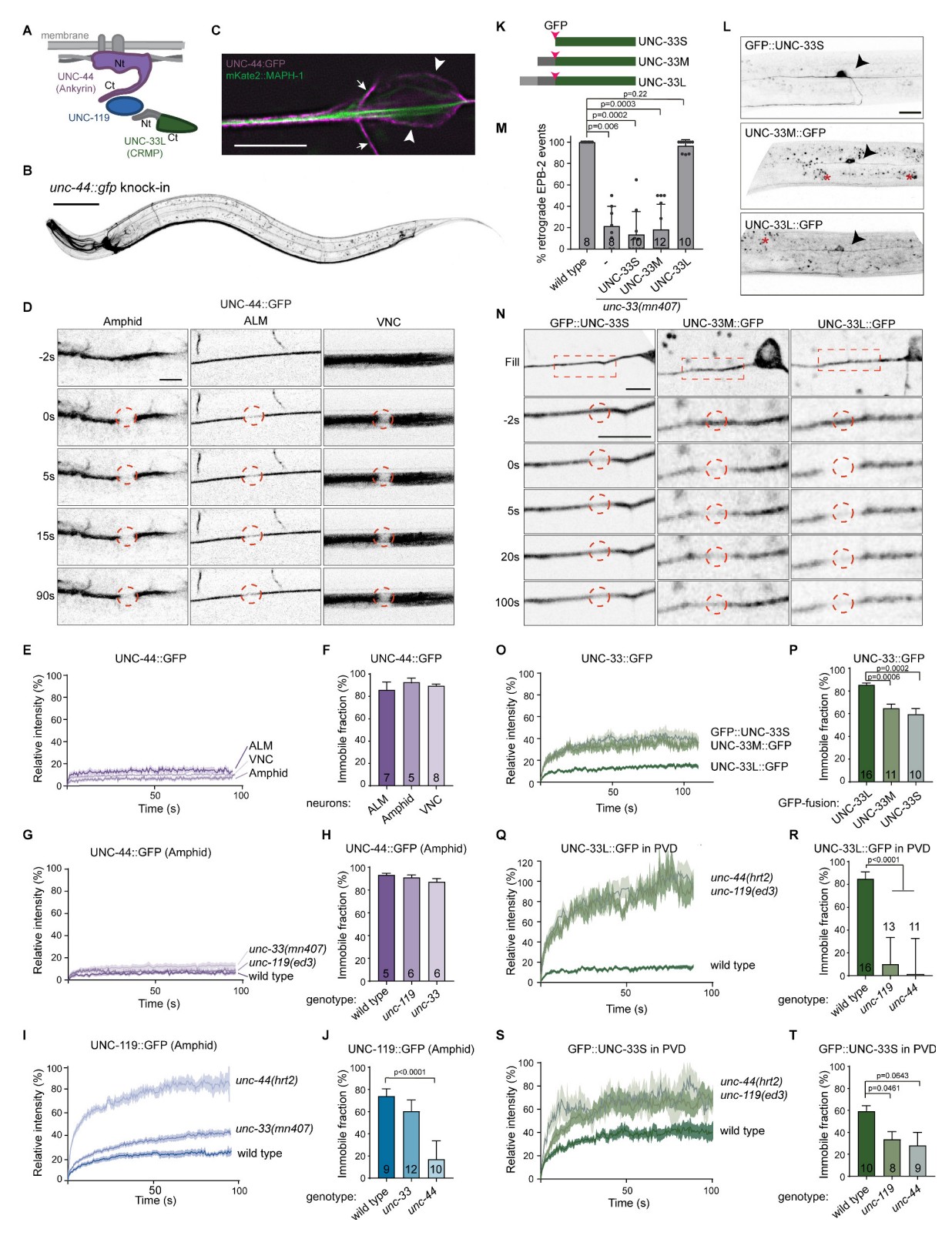

**Figure 3.** UNC-44 (Ankyrin) anchors UNC-33 (CRMP) via UNC-119. (**A**) Schematic representation of the UNC-44/UNC-119/UNC-33 complex (**B**) *unc-44::gfp* knock-in animals show neuronal GFP expression. The animals are superficially wild type, but the locomotion somewhat affected (***Figure 3—figure supplement 1A and B***). Scale, 50 μm. (**C**) Magnification of the ALM neuron cell body (arrowheads) and axon (extending to the left), expressing the microtubule marker mKate2::MAPH-1 (green) in the *unc-44::gfp* (magenta) knock-in animal. SRRF was used to improve the resolution

*Figure 3 continued on next page*

*Figure 3 continued*

(*Gustafsson et al., 2016*). The small arrows indicate a vertically crossing motor neuron axon. Scale, 5 μm. (D) UNC-44::GFP FRAP in the knock-in animal in amphid nerves (dendrites), the ALM axon or ventral nerve cord (VNC, mainly axons). Scale, 5 μm. (E–J) Average normalized intensity graphs of UNC-44::GFP (D,F) and UNC-119::GFP FRAP (H) ± SEM and the percentage immobile fractions after FRAP (E,G,I) in the indicated tissues and mutants. Number of analyzed animals is indicated. (K) Schematic representation of the three UNC-33 isoforms and GFP insertion. (L) Expression of GFP labeled UNC-33 isoforms in the PVD neuron. Red asterisk marks autofluorescent gut-granules. Scale, 20 μm. (M) Quantification of minus-end out microtubules in the PVD anterior dendrite in wild type or *unc-33* mutant with or without PVD specific expression of GFP labeled UNC-33 isoforms. (N) FRAP of GFP labeled UNC-33 isoforms in the PVD anterior dendrite. Scale, 5 μm. (O–T) Average normalized intensity graphs of UNC-33::GFP FRAP ± SEM (N,P,R) and the percentage immobile fractions after FRAP (O,Q,S) in the indicated tissues and mutants. Number of analyzed animals is indicated. All experiments analyzing GFP tagged UNC-44 and UNC-119 proteins were performed in the knock-in animals, whereas all experiments analyzing GFP tagged UNC-33 were done upon mild overexpression in the PVD neuron using single copy integrated strains (MosSCI; *Frøkjaer-Jensen et al., 2008*), except for the UNC-33M in (M). Analyzed animals were from the L4 or young adult stage; error bars represent SD; statistical analysis, Kruskal-Wallis test followed by Dunn's multiple comparisons test.

The online version of this article includes the following video, source data, and figure supplement(s) for figure 3:

**Source data 1.** Source data for graphs in *Figure 3*.
**Figure supplement 1.** UNC-44 (Ankyrin) anchors UNC-33 (CRMP) via UNC-119 supplement.
**Figure supplement 1—source data 1.** Source data for graphs in *Figure 3—figure supplement 1*.
**Figure 3—video 1.** FRAP microscopy of UNC-33L::GFP in the PVD neuron in wild type and *unc-44* mutant animals.
https://elifesciences.org/articles/55111#fig3video1

bind to UNC-119. Consistent with this, we observed the periodicity of UNC-33L, in striking similarity to UNC-119 and UNC-44 (*Figure 4C–E*; p=0.82; p=0.97; pairwise Wilcoxon rank sum test, respectively). In contrast, we did not observe a periodic pattern for the UNC-33M isoform (*Figure 4E*; p=4e-3, pairwise Wilcoxon rank sum test with UNC-119). The partial periodicity could be explained by the immobile fraction we also observed in the FRAP assay, that associates with microtubules. Taken together, the ternary UNC-44/UNC-119/UNC-33L complex shares a stereotypic periodic pattern similar to the cortical spectrin cytoskeleton.

## UNC-44 (Ankyrin) and UNC-119 anchor the microtubule binding UNC-33L (CRMP) to the cortex

CRMPs are described as cytosolic phosphoproteins that can bind microtubules via their C-terminus (*Lin et al., 2011*; *Niwa et al., 2017*; *Zheng et al., 2018*). To increase subcellular resolution, we ectopically expressed UNC-33S::GFP and UNC-33L::GFP in the hypodermal seam cells and found that both efficiently bind to microtubules and that the C-terminus is indeed essential for this interaction (*Figure 5A and B*; *Figure 5—video 1*). Interestingly, we observed that upon loss of microtubule binding, UNC-33L partially relocalizes to the cortex, whereas the shorter UNC-33S does not. In neurons, immobilization of UNC-33L depends on UNC-44 and UNC-119 (*Figure 3*) and although expression of these proteins is mainly neuronal, low levels are also present in the hypodermal seam cells (*Figure 1F*; *Chen et al., 2017*). Indeed, we found that the cortical relocalization of UNC-33L upon loss of microtubule binding depends on UNC-44 and UNC-119 (*Figure 5B*; *Figure 5—video 1*).

To address the dynamics of UNC-33 upon loss of microtubule binding in neurons, we expressed the three UNC-33 isoforms lacking the C-terminus (ΔC) in the PVD neuron (*Figure 5—figure supplement 1A*). With FRAP analysis, we found that UNC-33SΔC and UNC-33MΔC rapidly recover (*Figure 5C–G*), which indicates that the immobile pool of UNC-33S and UNC-33M (*Figure 3N-P*) is the result of microtubule binding. UNC-33LΔC however, is still highly immobile (*Figure 5B, H and I*). This suggests that although binding to microtubules via the C-terminus is essential for UNC-33L functioning (*Figure 5—figure supplement 1B*), it is not required for its immobilization. The UNC-33LΔC signal rapidly recovers in the *unc-119* mutant (*Figure 5J and K*), which shows that UNC-119 binds to the N-terminus to immobilize UNC-33L to UNC-44. In agreement, in the *unc-44* mutant UNC-33L appears to decorate the dynamic microtubules in the PVD cell body (*Figure 3—video 1*). Together, these results indicate that in neurons UNC-33L stably associates with microtubules via its C-terminus and to the cortex through its N-terminal interaction with UNC-119.

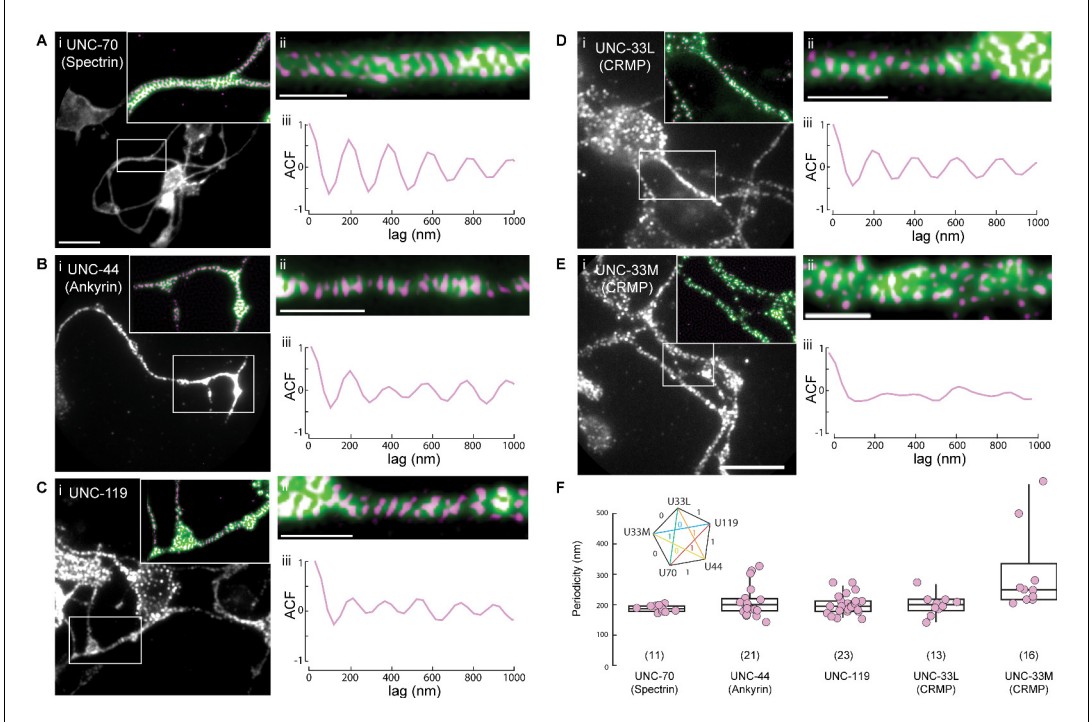

**Figure 4.** UNC-44 (Ankyrin), UNC-119 and UNC-33L (CRMP) form a periodic pattern along neurites. (A–E) Superresolution imaging of primary neurons from isolated *C elegans* cultures. Representative images of cells extracted from animals expressing GFP or GFP derivative tagged (A) UNC-70 (β-spectrin), (B) UNC-44, (C) UNC-119, (D) UNC-33L and (E) UNC-33M with the corresponding (i) widefield, (ii) supersolved images and (iii) autocorrelation function. The inset in (i) shows a magnified region with widefield (green) and superresolved pattern (magenta). Scale bars in (i) = 5 µm; in (ii) = 1 µm. (F) Quantification of the peak frequencies observed after Fourier decomposition of the line profiles, extracted from the superresolved images. ROIs were choses at various locations along the neurites showing periodicity with a minimum length of 2 µm. The inset shows the results from a non-parametric multicomparison test (Wilcoxon with Nemenyi post-hoc) for unequal sample size with $H_0$ that the median is from the same population. If test statistic $0.7 < q < 3.5$, $H_0 = 1$, otherwise $H_0 = 0$.

The online version of this article includes the following source data and figure supplement(s) for figure 4:

**Source data 1.** Source data for graphs in *Figure 4*.

**Figure supplement 1.** TIRF-SIM imaging setup and power spectral density plots.

## The UNC-44/119/33 complex immobilizes the neuronal microtubule cytoskeleton

The stability of the UNC-44/119/33 complex and the attachment of the microtubule binding UNC-33L (CRMP) to the cortex, suggests that the complex anchors microtubules in neurons. To visualize the microtubule cytoskeleton we expressed the microtubule binding protein MAPH-1 in the PVD neuron (*Harterink et al., 2018*) and found that microtubules are evenly distributed over the primary dendrites and the axon in wild type animals (*Figure 6A*). In the cell body, some microtubule mobility can be observed (*Figure 6—video 1*). Interestingly, in each of the *unc*-mutants we observed neuronal development defects and disorganization of the microtubule cytoskeleton; microtubules were no longer restricted to the primary dendrite, but penetrated the side branches as well (*Figure 6A and B*). Additionally, in these mutants microtubule mobility in the cell body is increased and we observed microtubule buckling at branchpoints (*Figure 6—video 1*). To examine the peculiar microtubule mobility, we constructed a TBA-1 (tubulin) with an N-terminal photoactivatable-GFP (PA-GFP) (*Kahn et al., 2018*; *Patterson and Lippincott-Schwartz, 2002*). Upon activating a small region in the dendrite, the cytoplasmic tubulin rapidly diffuses, while the fluorescent tubulin that is incorporated in the microtubule remains. In wild type worms, the photo-activated region hardly displaced over the course of several minutes (*Figure 6C–E*; *Figure 6—video 2*), which indicates that the neuronal microtubule cytoskeleton in a mature neuron is largely immobilized. Strikingly, this immobility was lost in the *unc-44* (Ankyrin), the *unc-119* and the *unc-33* (CRMP) mutants (*Figure 6C–*

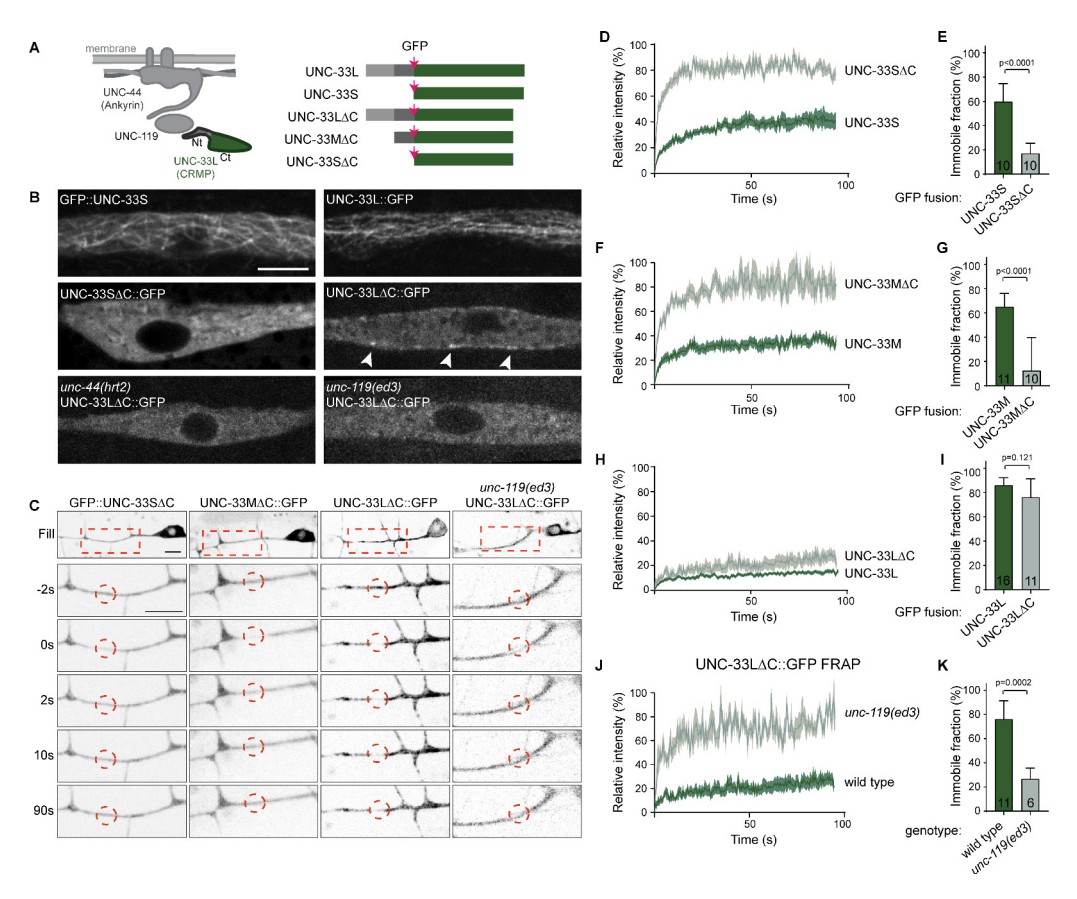

**Figure 5.** The microtubule binding UNC-33L (CRMP) is anchored to the cortex via its N-terminus. (**A**) Schematic representation of the UNC-44/UNC-119/UNC-33 complex and the different UNC-33 (CMRP) constructs used. (**B**) Ectopic expression of the indicated GFP tagged UNC-33 constructs in hypodermal seam cells. Arrowheads indicate cortical localized UNC-33LΔC::GFP. Scale, 5 μm. (**C**) FRAP imaging of the indicated GFP tagged UNC-33 constructs mildly overexpressed in the PVD neuron using single copy integrated strains (MosSCI; *Frøkjaer-Jensen et al., 2008*). (**D–K**) Average normalized intensity graphs of FRAP imaging of indicated GFP tagged UNC-33 constructs ± SEM (**D,F,H,J**) and the percentage immobile fractions after FRAP (**E,G,I,K**) in the indicated mutants. Number of analyzed animals is indicated. Analyzed animals were from the L4 or young adult stage; error bars represent SD; statistical analysis, unpaired t test.

The online version of this article includes the following video, source data, and figure supplement(s) for figure 5:

**Source data 1.** Source data for graphs in *Figure 5*.

**Figure supplement 1.** UNC-33L immobilization does not require microtubule binding.

**Figure supplement 1—source data 1.** Source data for graphs in *Figure 5—figure supplement 1*.

**Figure 5—video 1.** Expression of various GFP tagged UNC-33 constructs in the seam cells.

https://elifesciences.org/articles/55111#fig5video1

*E*; *Figure 6—video 2*) and we observed rapid transport of the photo-activated region with a bias towards the cell body: In the *unc*-mutants: in 12/20 animals the microtubules translocated towards the cell body and in 8/20 microtubules moved in both directions. The signal from the microtubule incorporated PA-GFP-TBA-1 was generally weaker in the mutants, which might suggest the presence of fewer microtubules in the PVD dendrite. Additionally, using the microtubules plus-end marker EBP-2::GFP we observed subtle changes in microtubule plus-end dynamics and growth event frequencies in the mutants (*Figure 6—figure supplement 1A–D*). These results show that the UNC-44/119/33 complex functions as a cortical anchoring complex to immobilize the microtubule cytoskeleton in neurons.

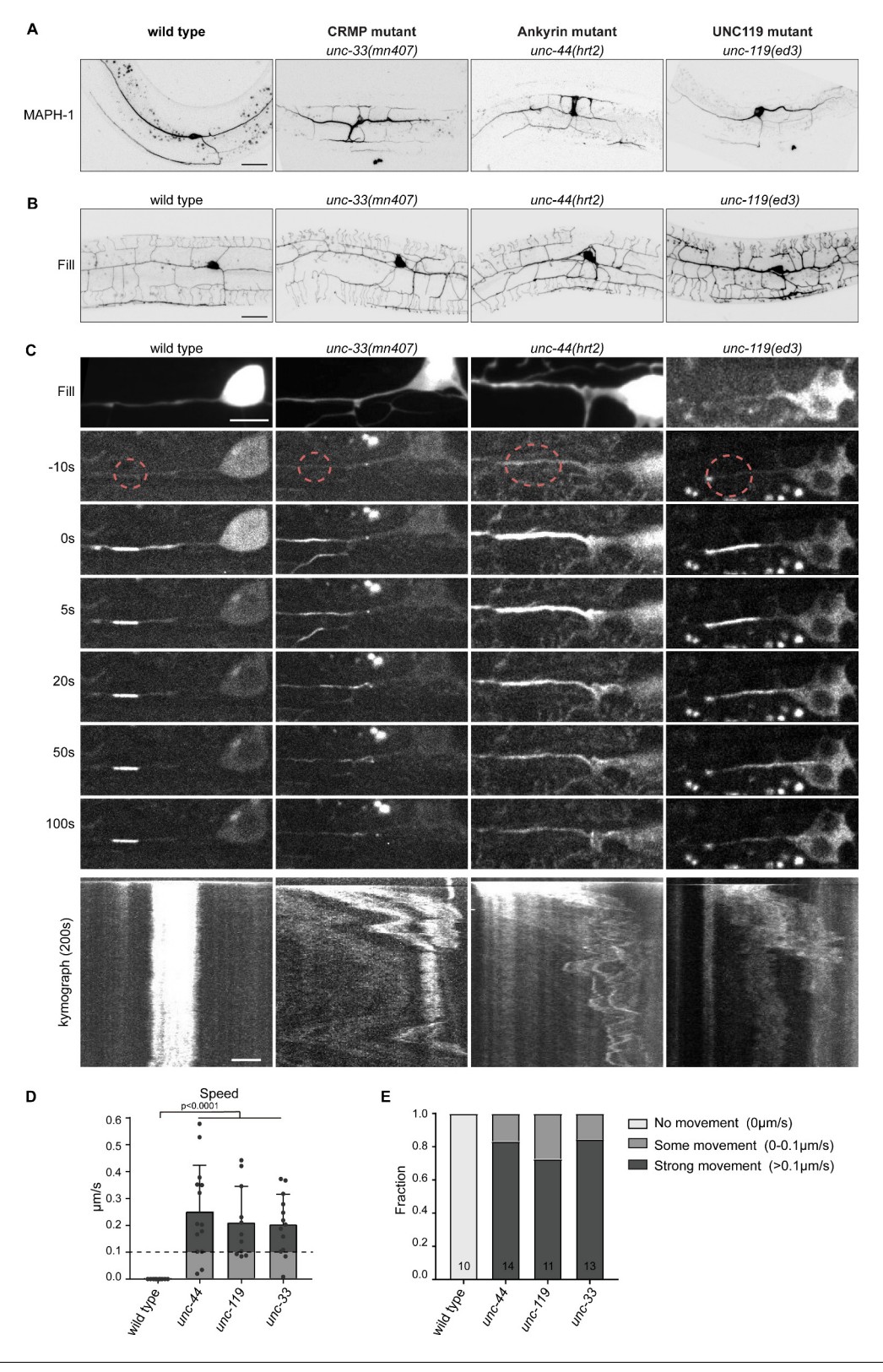

**Figure 6.** The cortical anchor complex immobilizes the microtubule cytoskeleton. (**A**) Representative images of mKate2::MAPH-1 microtubule marker in the PVD in indicated mutants. (see also **Figure 6—video 1**). Scale, 20 μm. (**B**) Representative images of PVD neuron morphology visualized with myristoylated GFP of the indicated mutants. Scale, 20 μm. (**C**) Representative stills and kymograph of microtubule mobility in the indicated mutants, visualized

*Figure 6 continued on next page*

*Figure 6 continued*

using PA-GFP::TBA-1 (tubulin) expressed in PVD. The photoactivated region is indicated. Scale, 5 µm for the stills and 2 µm for the kymographs. (**D–E**) Quantification of microtubule sliding. All sliding evens were averaged per animal. We defined an average sliding above 0.1 µm/s as strong sliding. Analyzed animals were from the L4 or young adult stage. Error bars represent SD; statistical analysis, unpaired student T-test.

The online version of this article includes the following video, source data, and figure supplement(s) for figure 6:

**Source data 1.** Source data for graphs in *Figure 6*.
**Figure supplement 1.** Microtubule plus-end dynamics in mutants for the cortical anchor complex.
**Figure supplement 1—source data 1.** Source data for graphs in *Figure 6—figure supplement 1*.
**Figure 6—video 1.** Visualization of the microtubule cytoskeleton using the microtubule binding protein MAPH-1, in the indicated genetic backgrounds.
https://elifesciences.org/articles/55111#fig6video1
**Figure 6—video 2.** Expression of PA-GFP::TBA-1 (tubulin) expressed in PVD neuron in indicated genetic backgrounds to visualize microtubule dynamics.
https://elifesciences.org/articles/55111#fig6video2

## The UNC-44/119/33 complex prevents kinesin-1 dependent microtubule sliding

The displacement of the microtubules in the *unc*-mutants suggests that forces are exerted on the microtubule cytoskeleton, potentially by microtubule motors. Previously, *unc-116* (kinesin-1) was shown to be important for dendritic microtubule organization, possibly by transporting microtubules (*Yan et al., 2013*). If the forces on microtubules are indeed generated by UNC-116, the *unc-116* mutant should not exhibit increased microtubule dynamics, since the microtubules are still anchored to the cortex by the UNC-44/119/33 complex. Furthermore, the microtubule sliding observed in *unc*-mutants should be suppressed by depletion of *unc-116*. We used an endogenously floxed *unc-116* mutant, which can knock-out *unc-116* specifically in the PVD neuron lineage (*Harterink et al., 2018*). As anticipated, we found that loss of *unc-116* did not induce microtubule mobility. More importantly, the depletion of *unc-116* (kinesin-1) completely suppressed the microtubule mobility in the *unc-33* (CRMP) mutant (*Figure 7A–C*; *Figure 7—video 1*), but not the microtubule polarity defect (*Figure 6—figure supplement 1D*) nor the morphology defects (not shown). Additionally, the partial loss-of-function mutant for the kinesin-1 adaptor, *klc-2* (kinesin light chain; KLC), also suppressed the microtubule mobility of the *unc-33* mutant (*Figure 7A–C*). This shows that in the absence of the cortical anchoring complex kinesin-1 slides microtubules through the neurites. This function appears specific to the kinesin-1 motor, as depletion of *unc-104* (kinesin-3) did not suppress the microtubule mobility in the *unc-33* mutant (*Figure 7A–C*; *Figure 7—video 1*).

## Direct coupling of UNC-33S (CRMP) to cortical UNC-44 (Ankyrin) rescues the *unc-119* mutant

Altogether we have found that UNC-119 bridges the membrane-associated UNC-44 (Ankyrin) to the microtubule binding UNC-33L (CRMP) to immobilize the microtubule cytoskeleton (*Figure 8A*). While in a wild type animal the UNC-116 (kinesin-1) motor is essential for neuronal polarity (*Yan et al., 2013*), in the absence of either of the components of the ternary complex, UNC-116 induces robust microtubule sliding (*Figure 7*). We propose that the UNC-44/119/33 complex acts as a membrane-associated anchoring complex to curb kinesin-1 motor activity for proper neuronal development.

To confirm this model, we examined whether artificial anchoring of UNC-33S (CRMP) is sufficient to balance kinesin-1 activity in the absence of the cortical anchoring complex. To do this, we fused UNC-33S, which cannot bind to UNC-119, to the GFP-nanobody (vhhGFP) and expressed it in the PVD neuron of *unc-44::gfp* knock-in animals (*Figure 8A*). Artificial coupling of UNC-33S to UNC-44 rescued the neuronal microtubule defects of the *unc-33* and *unc-119* mutants (*Figure 8B*) as well as the neuronal morphology of the *unc-33* mutant (*Figure 8C*). Remarkably, expression of the vhhGFP:: UNC-33S fusion protein in all neurons rescued the animal movement of the *unc-119* mutant (*Figure 8D and E* and *Figure 3—figure supplement 1B*; *Figure 8—video 1*), showing that neuronal

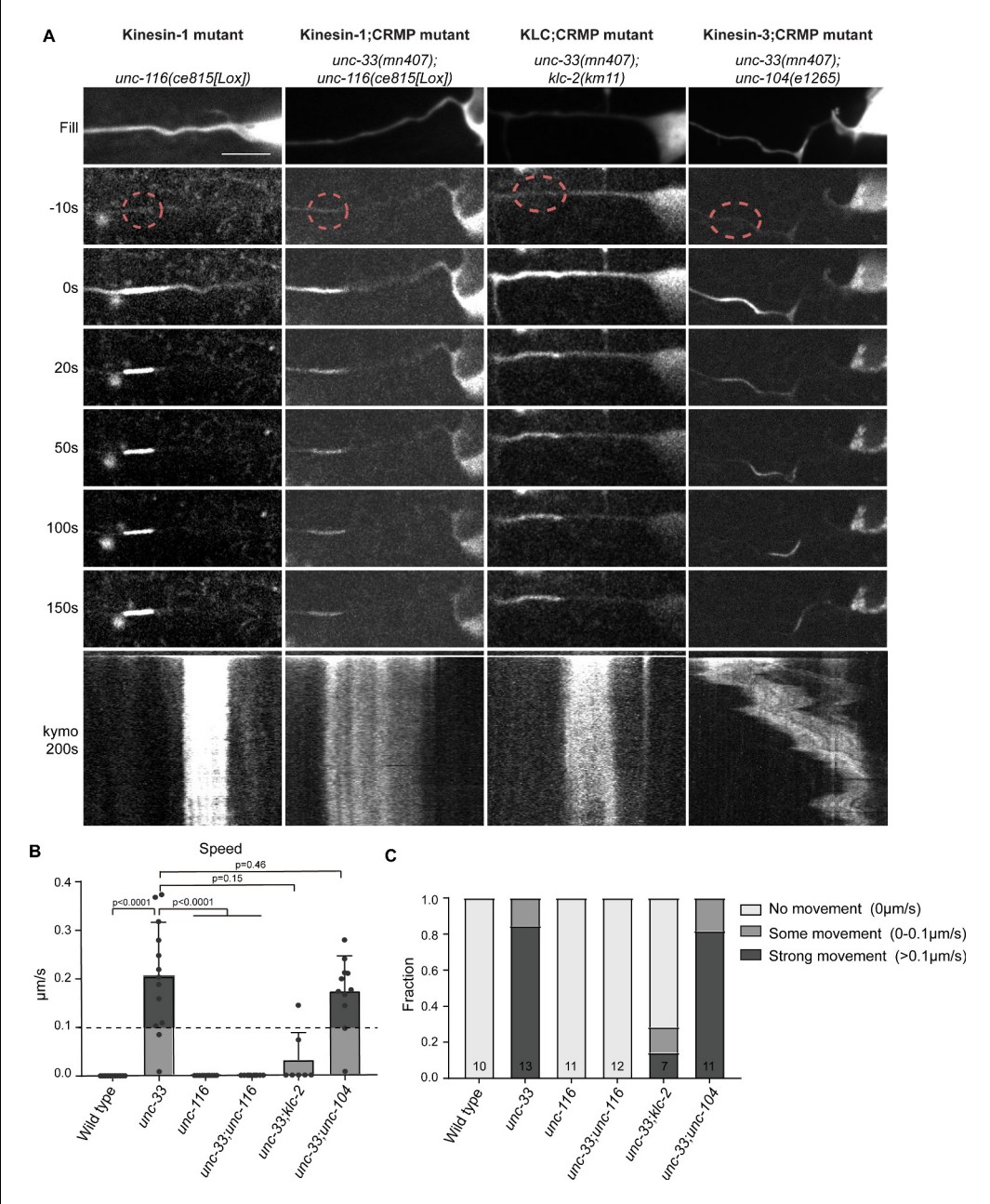

**Figure 7.** Kinesin-1 is sliding microtubule in the absence of the cortical anchor complex. (**A**) Representative stills and kymograph of microtubule mobility in the indicated mutants, visualized using PA-GFP::TBA-1 (tubulin) expressed in PVD. The photoactivated region is indicated. Scale, 5 μm for the stills and 2 μm for the kymographs. (**B–C**) Quantification of microtubule sliding. All sliding evens were averaged per animal. We defined an average sliding above 0.1 μm/s as strong sliding. Analyzed animals were from the L4 or young adult stage. Error bars represent SD; statistical analysis, unpaired student T-test.

The online version of this article includes the following video and source data for figure 7:

**Source data 1.** Source data for graphs in *Figure 7*.

**Figure 7—video 1.** Expression of PA-GFP::TBA-1 (tubulin) expressed in PVD neuron in indicated genetic backgrounds to visualize microtubule dynamics.

https://elifesciences.org/articles/55111#fig7video1

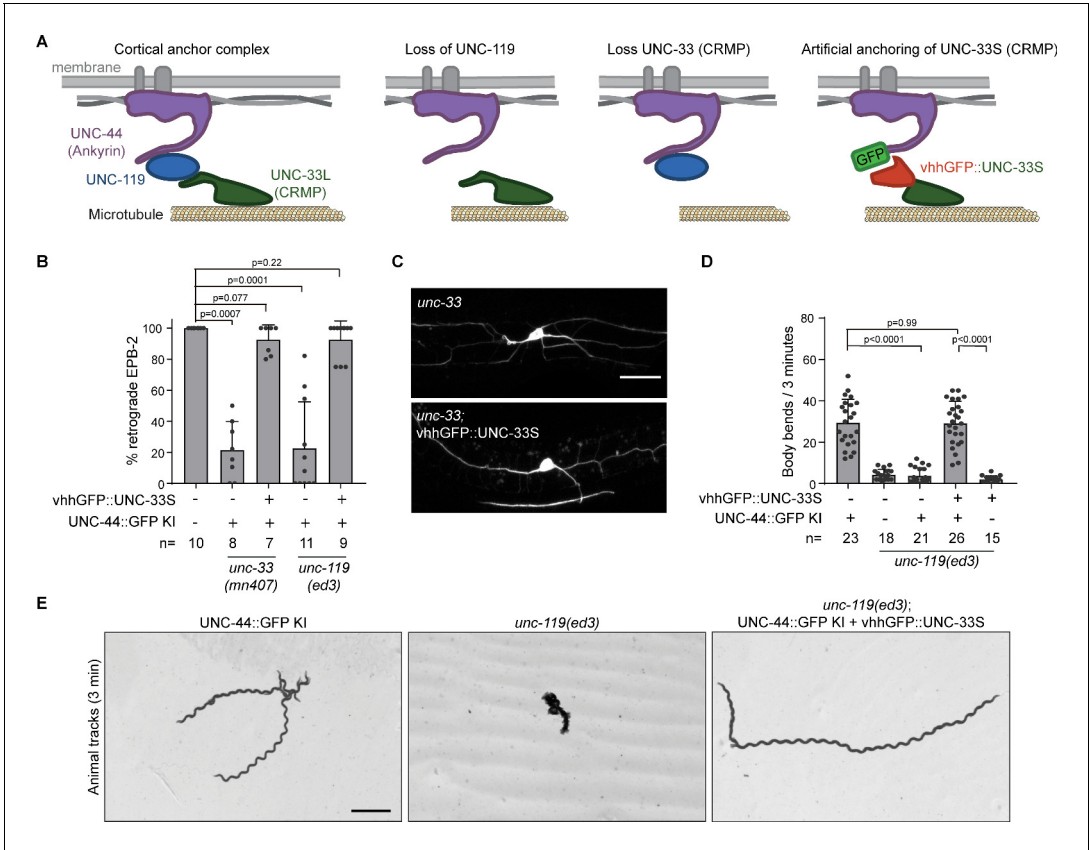

**Figure 8.** UNC-33 (CRMP) anchoring by UNC-119 is essential for neuron development and animal locomotion. (**A**) Model: The microtubule binding UNC-33L (CRMP) is cortically anchored to UNC-44 (Ankyrin) via UNC-119. In the absence of UNC-119 or the long UNC-33 isoform the putative microtubule binding UNC-33 is not anchored to the cortex. By fusing the UNC-33S (CRMP) to the GFP nanobody (vhhGFP) we can artificially anchor UNC-33 to the cortex by expressing the fusion protein in the *unc-44::gfp* knock-in animal. If the main function of UNC-119 neuron development is to anchor UNC-33L to the cortex, we expect that the artificial attachment of UNC-33 will suppress the *unc-119* and *unc-33* mutant phenotype. (**B–C**) Quantification of minus-end out microtubules in the PVD anterior dendrite (**B**) and examples of PVD neuron morphology (imaged with PVD >EBP-2:: mKate2) (**C**) with or without PVD specific expression of vhhGFP::UNC-33S together with the *unc-44::gfp* knock-in in indicated genetic backgrounds. Scale, 20 µm. (**D**) Quantification of the animal locomotion upon expression of pan-neuronal vhhGFP::UNC-33 in the indicated backgrounds. (**E**) Example tracks of indicated genotype during 3 min (maximum projection). Scale, 1 mm. Analyzed animals were from the L4 or young adult stage; error bars represent SD; statistical analysis, Kruskal-Wallis test followed by Dunn's multiple comparisons test.

The online version of this article includes the following video and source data for figure 8:

**Source data 1.** Source data for graphs in *Figure 8*.

**Figure 8—video 1.** Direct coupling of UNC-33 to UNC-44 by expressing vhhGFP::UNC-33S using a pan-neuronal Prab-3 promoter in the *unc-44::gfp* knock-in animals rescues animal locomotion (wild type is the *unc-44::gfp* knock-in).

https://elifesciences.org/articles/55111#fig8video1

functioning is restored and that the *unc-119* mutant locomotion defect is caused by a loss of UNC-33L anchoring to the cortical UNC-44 (Ankyrin) in neurons.

## Discussion

The *C. elegans unc-119* mutant is severely paralyzed, a phenotype that can easily be rescued by reintroduction of the gene and which has extensively been exploited as a transgenesis background in *C. elegans* for many year (*Maduro, 2015*). Although it is well known that *unc-119* is expressed in neurons (*Maduro and Pilgrim, 1995*) and to be important for neuron development (*Knobel et al., 2001*; *Materi and Pilgrim, 2005*), the molecular function underlying the severe paralysis of the mutant has remained elusive. Here we report that UNC-119 is part of a membrane anchored cytoskeleton complex that anchors the microtubule cytoskeleton to the membrane. Depletion of the

cortical anchoring complex leads to dramatic microtubule sliding induced by UNC-116 (kinesin-1). We expect that this excessive microtubule sliding is the underlying cause of the microtubule polarity defects and impaired neuronal development observed in the mutants. Since UNC-116 itself is essential for dendritic minus-end out microtubule organization (*Yan et al., 2013*), this suggests that neuronal development relies on a fine balance between microtubule sliding and microtubule immobilization by cortical anchoring.

A proper balance of microtubule sliding and bundle immobilization was also shown to be important for neuron development in other organisms. In *Drosophila*, microtubules are slid over other microtubules by kinesin-1 and potentially also by cortically anchored dynein, during the earliest phases of neurite outgrowth (*Del Castillo et al., 2015*). To accomplish this microtubule-microtubule sliding, kinesin-1 contains an additional microtubule binding site in its tail, which when mutated leads to developmental defects in both the axon and the dendrite, without affecting kinesin-1 cargo transport (*Jolly et al., 2010*; *Winding et al., 2016*). Mutating the sequence for the presumed extra microtubule binding site in the *C. elegans* UNC-116 motor tail reduced the capacity of the motor to organize microtubules minus-end out (*Yan et al., 2013*), arguing in favor of kinesin-1 driven microtubule-microtubule sliding. This kinesin-1 dependent microtubule sliding in *Drosophila* supposedly is independent of the kinesin light chain adaptors (*Jolly et al., 2010*; *Winding et al., 2016*). However, we found that, similar to depletion of UNC-116 (kinesin-1), depletion of KLC-2 also was able to suppress the microtubule sliding defect in the *unc-33* (CRMP) mutant. Future work will have to address the precise role of microtubule sliding by kinesin-1 in neuronal development.

UNC-119 is a highly conserved protein both in sequence and in function (*Maduro et al., 2000*). Interestingly, homologs for *unc-119* were found to be important for neuronal development in other species as well; depletion of zebrafish UNC-119a led to various neuronal defects that are reminiscent of the *C. elegans* phenotypes (*Manning et al., 2004*) and depletion of UNC-119A in cultured rat hippocampal neurons led to reduced dendritic branching (*May et al., 2014*). Thus far, the mechanism of these vertebrate UNC119 homologs in neuronal development has not been investigated at the molecular level. It is well documented that the mammalian UNC119 proteins can bind to the lipid moiety of myristoylated proteins (*Constantine et al., 2012*). As such, UNC119 acts as a chaperone to solubilize and transport the otherwise membrane-anchored proteins to other membranes or the cilium (*Konitsiotis et al., 2017*; *Wright et al., 2011*; *Zhang et al., 2011*), where the interaction with the GTPases Arl2 or Arl3 allosterically releases the myristoylated cargo from UNC119 (*Ismail et al., 2012*; *Kobayashi et al., 2003*). Similar functions have been proposed for UNC-119 *in C. elegans*. UNC-119 was shown to interact with Arl2/3 homologues, and the *unc-119* mutant has deficient transport of myristoylated proteins to the cilium as well as cilium formation defects (*Warburton-Pitt et al., 2014*; *Zhang et al., 2011*; *Zhang et al., 2016*). However, we observed no microtubule organization defects in mutants for the *C. elegans* Arl2/3 genes (results not shown). Moreover, we were able to rescue the *unc-119* mutant microtubule organization and locomotion by directly attaching UNC-33 to UNC-44, demonstrating that UNC-119 functions as a structural scaffold to bring these proteins together for proper neuronal development and functioning.

In *Drosophila*, the kinesin-1 sliding is counteracted by the 'mitotic' kinesin MKLP1 (*Del Castillo et al., 2015*), which potentially cross-links microtubules. In mammalian neurons, the stabilization of microtubule cytoskeleton relies on various MAPs that can bind and stabilize individual microtubules and cross-link them together (*Ramkumar et al., 2018*). This is especially apparent in the axon, where the microtubules are regularly spaced and accumulate modifications indicative for long lived microtubules (*Chen et al., 1992*; *Song and Brady, 2015*). Less is known, however, about how such a cross-linked microtubule array is immobilized itself in the neuron. In *Drosophila*, the microtubule cytoskeleton is stabilized by the spectraplakin protein Shot, which binds to both microtubules and the actin cytoskeleton (*Qu et al., 2017*; *Sanchez-Soriano et al., 2009*). Although the role of Shot in setting up axon-dendrite polarity is not addressed, in the absence of Shot, axon regeneration is affected and the microtubules are less bundled, which leads to buckling of microtubules. In *C. elegans* we observed similar microtubule buckling in the absence of the cortical anchoring complex (*Figure 6—video 1*). The dynamic association of the microtubule cytoskeleton to the spectrin-actin cytoskeleton is an important factor during neuron development reviewed by *Coles and Bradke (2015)*; *Krieg et al. (2017)*.

Neurons typically express giant Ankyrin isoforms that may have a common evolutionary origin and that have essential functions in neuron development (*Jegla et al., 2016*). Interestingly, these

giant Ankyrin proteins evolved several ways to connect the submembrane cytoskeleton to the microtubule network. In *Drosophila*, Ank2 was shown to directly bind to the microtubules (*Pielage et al., 2008*; *Stephan et al., 2015*), whereas in mammalian neurons, Ankyrin-G localizes to the axon initial segment, where it binds indirectly to microtubules via multiple EB proteins (*Fréal et al., 2016*; *Leterrier et al., 2011*). Recently it was found that the giant Ankyrin-B protein localizes to axons and that it is important for the microtubule organization in the axon (*Yang et al., 2019*). Since static enrichment of EB proteins is specific for the axon initial segment by binding to Ankyrin-G (*Leterrier et al., 2011*), it is unclear how Ankyrin-B connects to the microtubule cytoskeleton in the axon. Here, we show that in *C. elegans* the giant UNC-44 (Ankyrin) isoform interacts with microtubules via UNC-119 and UNC-33 (CRMP) in both axons and dendrites. Also in mammalian neurons, CRMPs are undeniably important for microtubule organization, neuronal development, and neuronal polarity (*Arimura et al., 2004*). This might suggest that the Ankyrin/UNC119/CRMP complex also functions as a microtubule anchor in other species. However, since UNC-119 binds to the non-conserved N-terminus of UNC-33L in *C. elegans*, further research is required to elucidate possible interactions and common functions of UNC-119 and UNC-33 in mammalian neurons.

In our experiments we observed that in mature neurons of wild type animals the microtubule cytoskeleton is for a large part immobile, as was seen before for other systems (*Cao et al., 2020*; *Kahn et al., 2018*; *Lu et al., 2013*). During neuronal development, however, growth and retraction of neurites is intimately linked to microtubule stability and dynamics (*Lewis et al., 2013*; *Rumpf et al., 2019*). Thus, it is likely that stability of the anchoring complex itself is regulated and/or that binding of the microtubules to the anchoring complex is regulated. We found that the direct coupling of UNC-33S to UNC-44 was able to bypass UNC-119 and thus to rescue the *unc-119* mutant phenotype. However, whether microtubules bind continuously to UNC-33 (CRMP), or whether their binding is regulated, for example by phosphorylation (*Zheng et al., 2018*), is not yet known. The coordination of microtubule sliding and anchoring during neuronal development will be an important focus for further research.

Interestingly it was recently shown that CRMPs are hyper-phosphorylated in the central nervous system of Alzheimer disease patients, and the phosphorylated CRMP was observed primarily in dystrophic neurons (*Mokhtar et al., 2018*). In addition, this amyloid beta-dependent CRMP phosphorylation is suggested to directly impair kinesin-1 association and thus axon development in humans. As also autism associated mutations in the giant Ankyrin-B gene were found to affect the microtubule organization in the axon (*Yang et al., 2019*), our observation of disorganized microtubule cytoskeleton in the absence of CRMP or Ankyrin, suggest a conserved mechanism of MT stabilization across evolution. Thus, understanding how the neuronal microtubule cytoskeleton is connected to other cellular structures not only has great potential to help understand neuronal development, but may also lead to new ways to treat damaged neurons.

# Materials and methods

## Key resources table

| Reagent type (species) or resource | Designation | Source or reference | Identifiers | Additional information |
|---|---|---|---|---|
| Antibody | Rabbit polyclonal anti-GFP | Abcam | Cat# ab290; RRID:AB_303395 | 1:10000 |
| Antibody | Mouse monoclonal anti-mCherry | Clontech Laboratories | Cat# 632543; RRID:AB_2307319 | 1:10000 |
| Antibody | Goat Anti-Rabbit IgG Antibody, IRDye 680LT Conjugated | LI-COR Biosciences | Cat# 827–11081; RRID:AB_10795015 | 1:3000 |
| Antibody | Goat Anti-Mouse IgG Antibody, IRDye 680LT Conjugated | LI-COR Biosciences | Cat# 827–11080; RRID:AB_10795014 | 1:3000 |
| Antibody | Goat Anti-Rabbit IgG Antibody, IRDye 800CW Conjugated | LI-COR Biosciences | Cat# 827–08365; RRID:AB_10796098 | 1:3000 |

*Continued on next page*

*Continued*

| Reagent type (species) or resource | Designation | Source or reference | Identifiers | Additional information |
|---|---|---|---|---|
| Antibody | Goat Anti-Mouse IgG Antibody, IRDye 800CW Conjugated | LI-COR Biosciences | Cat# 827–08364; RRID:AB_10793856 | 1:3000 |
| Cell line | Human embryonic kidney 239T (HEK293T) | ATCC | CRL-3216 RRID:CVCL_0063 | |
| Chemicals | PEI | PolySciences | Cat# 24765–2 | |
| Software | ImageJ | NIH | https://imagej.nih.gov/ij/; RRID:SCR_003070 | |
| Software | GraphPad Prism 8 | GraphPad | https://www.graphpad.com/scientific-software/prism/; RRID:SCR_002798 | |
| Software | Kymoreslicewide | Github | https://github.com/ekatrukha/KymoResliceWide | |
| Strain, strain background (*C. elegans*) | *unc-33(mn407); kyIs445[des-2::mCherry:: rab-3; des-2::sad-1::gfp; Podr-1::dsRed]* | Cross - this paper | STR110 | |
| Strain, strain background (*C. elegans*) | *unc-119(ed3); hrtIs3 [Pdes-2::myr::GFP; Punc-122::dsRed]* | Cross - this paper | STR174 | |
| Strain, strain background (*C. elegans*) | *unc-44(tm349);hrtIs3 [Pdes-2::myristoyl::GFP, Punc-122::DsRed]* | Cross - this paper | STR176 | |
| Strain *Caenorhabditis elegans* | *unc-44(hrt2)* | *Harterink et al., 2018* | STR237 | |
| Strain, strain background (*C. elegans*) | *unc-44(hrt5[GFP-KI])* | Injected - this paper | STR282 | strain was generated using *pha-1* co-CRISPR |
| Strain, strain background (*C. elegans*) | *unc-33(mn407); unc-44(hrt5[GFP-KI])* | Cross - this paper | STR292 | |
| Strain, strain background (*C. elegans*) | *unc-119(ed3); hrtSi26 [Pdes-2::ebp-2::mKate2 LGII]* | Injected - this study | STR316 | |
| Strain, strain background (*C. elegans*) | *unc-119(ed3); hrtSi26 [Pdes-2::ebp-2::mKate2 LGII]* | Injected - this paper | STR316 | |
| Strain, strain background (*C. elegans*) | *unc-119(ed3); hrtSi28 [Pdes-2::mKate2::maph-1.1 LGIV]* | *Harterink et al., 2018* | STR318 | |
| Strain, strain background (*C. elegans*) | *hrtEx127[Punc-86::ebp-2:: gfp; Pmyo-2::tdTomato]* | Injected - this paper | STR366 | |
| Strain, strain background (*C. elegans*) | *unc-33(hrt7); unc-119(ed3); hrtSi28[Pdes-2::mKate2:: maph-1.1 LGIV]* | Injected - this paper | STR367 | *hrt7* was generated using CRISPR (11 bps deletion) |
| Strain, strain background (*C. elegans*) | *unc-119(ed3); hrtSi41 [Pdes-2::unc-33L::gfp LGI]* | Injected - this paper | STR369 | *gfp* is inserted in *unc-33* at the start of the S isoform |

*Continued on next page*

Continued

| Reagent type (species) or resource | Designation | Source or reference | Identifiers | Additional information |
|---|---|---|---|---|
| Strain, strain background (C. elegans) | unc-33(mn407); kyIs445 [des-2::mCherry::rab-3; des-2::sad-1::gfp; Podr-1::dsRed]; hrtSi41 [Pdes-2::unc-33L::gfp LGI] | Cross - this paper | STR386 | |
| Strain, strain background (C. elegans) | unc-44(hrt2); hrtSi41 [Pdes-2::unc-33L::gfp LGI] | Cross - this paper | STR387 | |
| Strain, strain background (C. elegans) | unc-44(hrt8); unc-119(ed3); hrtSi28[Pdes-2::mKate2:: maph-1.1 LGIV] | Injected - this study | STR388 | hrt8 was generated using CRISPR (11 bps insertion) |
| Strain, strain background (C. elegans) | unc-33(mn407); hrtSi26 [Pdes-2::ebp-2::mKate2 LGII]; hrtSi41[Pdes-2::unc-33L::gfp LGI] | Cross - this paper | STR405 | |
| Strain, strain background (C. elegans) | unc-119(ed3); hrtSi54 [Pdes-2::unc-33M::gfp LGI] | Injected - this paper | STR425 | gfp is inserted in unc-33 at the start of the S isoform |
| Strain, strain background (C. elegans) | unc-33(mn407); hrtSi26 [Pdes-2::ebp-2::mKate2 LGII] | Cross - this paper | STR431 | |
| Strain, strain background (C. elegans) | unc-33(mn407); hrtSi26 [Pdes-2::ebp-2::mKate2 LGII] | Cross - this paper | STR431 | |
| Strain, strain background (C. elegans) | unc-119(ed3); unc-44(hrt5[GFP-KI]) | Cross - this paper | STR434 | |
| Strain, strain background (C. elegans) | unc-44(hrt5[GFP-KI]); hrtSi50 [Pmec-4::mKate2::maph-1.1 LGIV] | Injected and Cross - this paper | STR439 | |
| Strain, strain background (C. elegans) | unc-119(ed3); wyEx4828 [Pdes-2::ebp-2::gfp; Podr- 1::RFP] | Cross - this paper | STR443 | |
| Strain, strain background (C. elegans) | unc-33(mn407); kyIs445 [des-2::mCherry::rab-3; des-2::sad-1::gfp; Podr-1::dsRed]; hrtSi54[Pdes-2::unc-33M::gfp LGI] | Cross - this paper | STR444 | |
| Strain, strain background (C. elegans) | unc-119(ed3); hrtEx142 [Pdes-2::unc-119::gfp; Pdes-2:: ebp-2::mKate2; Pmyo-2::tdTomato] | Injected - this paper | STR448 | |
| Strain, strain background (C. elegans) | unc-119(ed3); hrtEx143 [Pdes-2::ebp-2::mKate2; Pmyo-2::tdTomato (no cb-unc-119)] | Injected - this paper | STR449 | |
| Strain, strain background (C. elegans) | unc-119(hrt13[GFP-KI]) | Injected - this paper | STR485 | |
| Strain, strain background (C. elegans) | unc-33(mn407); unc-119(hrt13[GFP-KI]) | Cross - this paper | STR499 | |
| Strain, strain background (C. elegans) | unc-44(hrt2); unc-119(hrt13[GFP-KI]) | Cross - this paper | STR500 | |
| Strain, strain background (C. elegans) | unc-119(ed3); hrtSi69[Pdes-2:: gfp::unc-33S LGI] | Injected - this paper | STR512 | |

*Continued*

| Reagent type (species) or resource | Designation | Source or reference | Identifiers | Additional information |
|---|---|---|---|---|
| Strain, strain background (*C. elegans*) | unc-33(mn407); hrtSi26[Pdes-2::ebp-2::mKate2 LGII]; hrtSi69[Pdes-2:: gfp::unc-33S LGI] | Cross - this paper | STR532 | |
| Strain, strain background (*C. elegans*) | unc-44(hrt5[GFP-KI]); hrtSi50[Pmec-4::mKate2:: maph-1.1 LGIV] | Injected and Cross - this paper | STR439 | |
| Strain, strain background (*C. elegans*) | unc-119(ed3); wyEx4828[Pdes-2::ebp-2:: gfp; Podr- 1::RFP] | Cross - this paper | STR443 | |
| Strain, strain background (*C. elegans*) | unc-33(mn407); kyIs445[des-2::mCherry::rab-3; des-2::sad-1::gfp; Podr-1::dsRed]; hrtSi54[Pdes-2::unc-33M::gfp LGI] | Cross - this paper | STR444 | |
| Strain, strain background (*C. elegans*) | unc-119(ed3); hrtEx142[Pdes-2::unc-119::gfp; Pdes-2::ebp-2::mKate2; Pmyo-2::tdTomato] | Injected - this paper | STR448 | |
| Strain, strain background (*C. elegans*) | unc-119(ed3); hrtEx143 [Pdes-2::ebp-2::mKate2; Pmyo-2:: tdTomato (no cb-unc-119)] | Injected - this study | STR449 | |
| Strain, strain background (*C. elegans*) | unc-119(hrt13[GFP-KI]) | Injected - this paper | STR485 | |
| Strain, strain background (*C. elegans*) | unc-33(mn407); unc-119(hrt13[GFP-KI]) | Cross - this paper | STR499 | |
| Strain, strain background (*C. elegans*) | unc-44(hrt2); unc-119(hrt13[GFP-KI]) | Cross - this paper | STR500 | |
| Strain, strain background (*C. elegans*) | unc-119(ed3); hrtSi69[Pdes-2:: gfp::unc-33S LGI] | Injected - this paper | STR512 | |
| Strain, strain background (*C. elegans*) | unc-33(mn407); hrtSi26[Pdes-2::ebp-2:: mKate2 LGII]; hrtSi69 [Pdes-2::gfp::unc-33S LGI] | Cross - this paper | STR532 | |
| Strain, strain background (*C. elegans*) | unc-44(hrt5[GFP-KI]); hrtSi50[Pmec-4::mKate2: :maph-1.1 LGIV] | Injected and Cross - this paper | STR439 | |
| Strain, strain background (*C. elegans*) | unc-119(ed3); wyEx4828[Pdes-2:: ebp-2::gfp; Podr- 1::RFP] | Cross - this paper | STR443 | |
| Strain, strain background (*C. elegans*) | unc-33(mn407); kyIs445[des-2::mCherry::rab-3; des-2::sad-1::gfp; Podr-1::dsRed]; hrtSi54[Pdes-2::unc-33M::gfp LGI] | Cross - this paper | STR444 | |
| Strain, strain background (*C. elegans*) | unc-119(ed3); hrtEx142[Pdes-2::unc-119::gfp; Pdes-2::ebp-2::mKate2; Pmyo-2::tdTomato] | Injected - this paper | STR448 | |
| Strain, strain background (*C. elegans*) | unc-119(ed3); hrtEx143 [Pdes-2::ebp-2::mKate2; Pmyo-2::tdTomato (no cb-unc-119)] | Injected - this paper | STR449 | |
| Strain, strain background (*C. elegans*) | unc-119(hrt13[GFP-KI]) | Injected - this paper | STR485 | |

*Continued*

| Reagent type (species) or resource | Designation | Source or reference | Identifiers | Additional information |
|---|---|---|---|---|
| Strain, strain background (*C. elegans*) | *unc-33(mn407); unc-119 (hrt13[GFP-KI])* | Cross - this paper | STR499 | |
| Strain, strain background (*C. elegans*) | *unc-44(hrt2); unc-119 (hrt13[GFP-KI])* | Cross - this paper | STR500 | |
| Strain, strain background (*C. elegans*) | *unc-119(ed3); hrtSi69 [Pdes-2::gfp::unc-33S LGI]* | Injected - this study | STR512 | |
| Strain, strain background (*C. elegans*) | *unc-33(mn407); hrtSi26[Pdes-2::ebp-2:: mKate2 LGII]; hrtSi69[Pdes-2:: gfp::unc-33S LGI]* | Cross - this paper | STR532 | |
| Strain, strain background (*C. elegans*) | *hrtEx161[Pdes-2:: PA-GFP::tba-1; Pdes-2:: mKate2; Pmyo-2::mCherry]* | Injected - this paper | STR536 | |
| Strain, strain background (*C. elegans*) | *unc-119(ed3); hrtEx165 [Pwrt-2::unc-33L::gfp]* | Injected - this paper | STR544 | *gfp* is inserted in *unc-33* at the start of the S isoform |
| Strain, strain background (*C. elegans*) | *unc-119(ed3); hrtEx166 [Pwrt-2::gfp::unc-33S]* | Injected - this paper | STR545 | |
| Strain, strain background (*C. elegans*) | *unc-119(ed3); hrtSi70 [Pdes-2::unc-33LΔC::gfp LGI]* | Injected - this paper | STR546 | *gfp* is inserted in *unc-33* at the start of the S isoform |
| Strain, strain background (*C. elegans*) | *unc-44(hrt2); hrtEx161 [Pdes-2::PA-GFP::tba-1; Pdes-2::mKate2; Pmyo-2::mCherry]* | Cross - this paper | STR548 | |
| Strain, strain background (*C. elegans*) | *unc-119(ed3); hrtEx167 [Pwrt-2::gfp::unc-33SΔC]* | Injected - this paper | STR550 | *gfp* is inserted in *unc-33* at the start of the S isoform |
| Strain, strain background (*C. elegans*) | *unc-33(mn407); hrtSi26[Pdes-2::ebp-2::mKate2 LGII]; hrtEx168[Pdes-2:: unc-33M::gfp;Pmyo-2::mCherry]* | Injected - this paper | STR552 | |
| Strain, strain background (*C. elegans*) | *unc-119(ed3); hrtSi73 [Prab-3::unc-33M::gfp]* | Injected - this paper | STR553 | *gfp* is inserted in *unc-33* at the start of the S isoform |
| Strain, strain background (*C. elegans*) | *unc-119(ed3); hrtSi74 [Prab-3::unc −33L::gfp]* | Injected - this paper | STR554 | *gfp* is inserted in *unc-33* at the start of the S isoform |
| Strain, strain background (*C. elegans*) | *unc-33(mn407); hrtSi26[Pdes-2::ebp-2:: mKate2 LGII]; hrtSi70 [Pdes-2::gfp::unc33LΔC LGI]* | Cross - this paper | STR559 | |
| Strain, strain background (*C. elegans*) | *unc-119(ed3);hrtSi75 [Pdes-2::unc-33MΔC::gfp LGI]* | Injected - this paper | STR561 | *gfp* is inserted in *unc-33* at the start of the S isoform |
| Strain, strain background (*C. elegans*) | *unc-33(mn407); hrtEx161[Pdes-2::PA-GFP:: tba-1; Pdes-2::mKate2; Pmyo-2::mCherry]* | Cross - this paper | STR563 | |
| Strain, strain background (*C. elegans*) | *unc-116(ce815[LoxP1/unc-116/sup-1/LoxP2]); heSi175 [Pscm::CRE]; hrtEx161[Pdes-2:: PA-GFP::tba-1; Pdes-2::mKate2; Pmyo-2::mCherry]* | Cross - this paper | STR564 | |

*Continued on next page*

*Continued*

| Reagent type (species) or resource | Designation | Source or reference | Identifiers | Additional information |
|---|---|---|---|---|
| Strain, strain background (*C. elegans*) | *unc-33(mn407); unc-116(ce815 [LoxP1/unc-116/sup-1/LoxP2]); heSi175[Pscm::CRE]; hrtEx161 [Pdes-2::PA-GFP::tba-1; Pdes-2:: mKate2; Pmyo-2::mCherry]* | Cross - this paper | STR575 | |
| Strain, strain background (*C. elegans*) | *unc-33(mn407); unc-44(hrt5[GFPKI]); hrtSi26[Pdes-2::ebp-2::mKate2 LGII]; hrtEx173[Pdes-2::BFP-NLS::p2A: :vhhGFP::unc-33S::tbb-2UTR; Pmyo-2::mCherry]* | Injected - this paper | STR576 | |
| Strain, strain background (*C. elegans*) | *hrtIs3[Pdes-2::myr-GFP; Punc-122::dsRed]* | *Harterink et al., 2018* | STR58 | |
| Strain, strain background (*C. elegans*) | *unc-44(hrt2); hrtEx175 [Pwrt-2::unc33LΔC::gfp; Pmyo-2::mCherry]* | Injected - this paper | STR584 | |
| Strain, strain background (*C. elegans*) | *hrtEx175[Pwrt-2::unc33LΔC:: gfp; Pmyo-2::mCherry]* | Cross - this paper | STR585 | *gfp* is inserted in *unc-33* at the start of the S isoform |
| Strain, strain background (*C. elegans*) | *unc-119(ed3); hrtSi81 [Pdes-2::gfp::unc33SΔC LGI]* | Injected - this paper | STR588 | |
| Strain, strain background (*C. elegans*) | *unc-33(mn407); unc-44(hrt5 [GFPKI]); hrtSi26[Pdes-2:: ebp-2::mKate2 LGII]* | Cross - this study | STR591 | |
| Strain, strain background (*C. elegans*) | *unc-33(mn407); unc-104(e1265); hrtEx161[Pdes-2::PA-GFP::tba-1; Pdes-2::mKate2;Pmyo-2::mCherry]* | Cross - this paper | STR592 | |
| Strain, strain background (*C. elegans*) | *klc-2(km11); unc-33(mn407); hrtEx161[Pdes-2::PA-GFP::tba-1; Pdes-2::mKate2;Pmyo-2::mCherry]* | Cross - this paper | STR594 | |
| Strain, strain background (*C. elegans*) | *unc-119(ed3);hrtEx178[Pdes-2:: PA-gfp::tba-1::tbb-2UTR; Pdes-2::bfp; Pmyo-2::mCherry (no cb-unc-119)]* | Injected - this paper | STR595 | |
| Strain, strain background (*C. elegans*) | *unc-119(ed3); hrtSi87[Pdes-2:: mKate2::maph-1.1 (no cb-unc-119) LGIV]* | Injected - this paper | STR601 | *hrtSi87* was generated using CRISPR to mutate *cb-unc-119* in *hrtSi28* (17 bps deletion) |
| Strain, strain background (*C. elegans*) | *unc-119(ed3); hrtS86[Pdes-2::unc-33L::gfp (no cb-unc-119) LGI]* | Injected - this paper | STR608 | *gfp* is inserted in *unc-33* at the start of the S isoform; *hrtSi86* was generated using CRISPR to mutate *cb-unc-119* in *hrtSi41* (2 bp deletion and 42 bp insertion) |
| Strain, strain background (*C. elegans*) | *unc-119(ed3); unc-44(hrt5[GFPKI]; hrtEx181[Prab-3::BFP-NLS::p2A:: vhhGFP::unc-33s;Pmyo-2:: mCherry (no cb-unc-119)]* | Injected - this paper | STR619 | |
| Strain, strain background (*C. elegans*) | *unc-119(ed3); hrtSi89[Pdes-2::ebp-2::gfp (no cb-unc-119) LGI]* | Injected - this paper | STR620 | *hrtSi89* was generated using CRISPR to mutate *cb-unc-119* in *hrtSi5* (10 bps deletion) |
| Strain, strain background (*C. elegans*) | *unc-119; hrtSi90[Pdes-2::unc-33LdeltaC::GFP (no cb-unc-119)]* | Injected - this paper | STR621 | *gfp* is inserted in *unc-33* at the start of the S isoform; *hrtSi90* was generated using CRISPR to mutate *cb-unc-119* in *hrtSi70* (seven bps deletion) |

*Continued on next page*

*Continued*

| Reagent type (species) or resource | Designation | Source or reference | Identifiers | Additional information |
|---|---|---|---|---|
| Strain, strain background (*C. elegans*) | unc119(ed3); hrtSi91[[Pgcy-36::ebp-2::gfp(no cb-unc-119)] | Injected - this paper | STR622 | *hrtSi91* was generated using CRISPR to mutate *cb-unc-119* in *hrtSi4* (10pbs deletion) |
| Strain, strain background (*C. elegans*) | unc-119(ed3); hrtSi92[Pdes-2::gfp::unc-33S (no cb-unc-119) LGI] | Injected - this study | STR623 | *hrtSi92* was generated using CRISPR to mutate *cb-unc-119* in *hrtSi69* (10pbs deletion) |
| Strain, strain background (*C. elegans*) | unc-119(ed3); unc-44(hrt5[GFPKI]); hrtEx182[Pdes-2::BFP-NLS::p2A::vhhGFP::unc-33S;Pdes-2::ebp-2::mKate2;Pmyo-2::mCherry (no cb-unc-119)] | Injected - this paper | STR624 | |
| Strain, strain background (*C. elegans*) | unc-44(hrt2); hrtSi69 [Pdes-2::gfp::unc-33S LGI] | Cross - this paper | STR625 | |
| Strain, strain background (*C. elegans*) | unc-119(ed3);hrtEx186 [Pwrt2::unc-33LΔC::gfp (no cb-unc-119)] | Injected - this paper | STR634 | *gfp* is inserted in *unc-33* at the start of the S isoform |
| Strain, strain background (*C. elegans*) | unc-119(ed3); hrtEx181 [Prab-3::BFP-NLS::p2A::vhhGFP::unc-33s;Pmyo-2::mCherry (no cb-unc-119)] | Cross - this paper | STR635 | |
| Strain, strain background (*C. elegans*) | hrtSi4[Pgcy-36::ebp-2::gfp LGI] | *Harterink et al., 2018* | STR66 | |
| Strain, strain background (*C. elegans*) | hrtSi5[Pdes-2::ebp-2::gfp LGI] | *Harterink et al., 2018* | STR71 | |
| Strain, strain background (*C. elegans*) | unc-33(mn407); hrtIs3 [Pdes-2::myristoyl::GFP, Punc-122::DsRed] | Cross - this paper | STR95 | |
| Strain, strain background (*C. elegans*) | kyIs445[des-2::mCherry::rab-3; des-2::sad-1::gfp; Podr-1::dsRed] | *Maniar et al., 2012* | CX9797 | |
| Strain, strain background (*C. elegans*) | wyEx4828[Pdes-2::ebp-2::gfp; Podr- 1::RFP] | *Yan et al., 2013* | TV11781 | |
| Strain, strain background (*C. elegans*) | unc-33(mn407); unc-116 (ce815[LoxP1/unc-116/sup-1/LoxP2]); heSi175[Pscm::CRE];hrtSi5 | Cross - this paper | STR615 | |
| Strain, strain background (*C. elegans*) | pgIs22 [unc-70::N-TSmod]. oxIs95 [pdi-2p::unc-70 + myo-2p::GFP] | *Krieg et al., 2017* | GN600 | |
| Plasmid | sgRNA targeting sequence to generate the unc-119-GFP knock-in | This paper | pMH645 | GATGCATAATTTCCCGCCGA |
| plasmid | sgRNA targeting sequence to generate the unc-44-GFP knock in hrt8 knock-out | This paper | pMH243 | GACACGTATGAATCCGCCCA |
| plasmid | sgRNA targeting sequence to generate the unc-33(hrt7) mutant | This paper | pMH416 | GATGTCGTCGGCAATGATGG |
| RNA oligos | sgRNA targeting sequence to generate mutate cb-unc-119 in mosSCI lines | IDT | unc-119 CB (RNA guide) | CCTTGTTCGGTGCTTGGTGG |

### *C. elegans* culturing, constructs and transgenesis

Strains were cultured using standard conditions at 15°C or 20°C and imaged at room temperature. All strains used for this study can be found in the key recourse table and if generated in this study are available upon request. New transgenic strains were generated by injection of the indicated constructs forming extra-chromosomal lines, single copy insertions using mosSCI (*Frøkjaer-Jensen et al., 2008*) or by endogenous insertions/deletions using CRISPR. *unc-33(hrt7)*, *unc-44(hrt8)* were generated by injection of the guide + Cas9 plasmid + Pmyo-2::tdTomato into STR318 and screening for Unc phenotype. Similarly, the deletion of *cb-unc-119* form mosSCI lines was performed by injecting *cb-unc-119* sgRNA with Cas9 protein and scoring for the Unc phenotype. To generate *unc-44(hrt5[GFP])* we used *pha-1* co-conversion approach (*Ward, 2015*) and to generate *unc-119 (hrt13[GFP])* we used the self-excising cassette-approach (*Dickinson et al., 2015*). Constructs were generated using multisite Gateway cloning (Invitrogen) or by Gibson assembly using NEBuilder (New England Biolabs) and were sequenced for verification (all cloning details can be found in *Supplementary file 2*).

## Locomotion assay

Pick 10–20 L4 animals from an unstarved plate to fresh plates, one worm to a plate, and put all the plates in room temperature for 10 min. Gently place the plate on the dissecting scope. Start a 3 min timer, and begin counting body bends. If an animal stopped moving or reversed direction, the count was abandoned. If the animal is at the edge of the lawn, put the plate aside for a few minutes and come back to it for measurement when the worm is away from the lawn edge. Count the all the worms in the same day, using the same batch of plates.

### Primary cell culture and immunocytochemistry

## Primary cell culture

Embryonic cell isolation was performed using a previously described method with some modifications (*Krieg et al., 2017*). Briefly, synchronized worms were seeded onto peptone enriched plates and incubated at RT until the plates were populated with eggs. Then, the plates were washed off and the eggs were collected using milli-Q $H_2O$. The worm and egg pellets were resuspended in the freshly prepared bleaching solution and rocked gently for 4–7 min, once 70–80% of the worms were lysed the reaction was stopped using egg buffer (118 mM NaCl, 48 mM KCl, 2 mM $CaCl_2$, 2 mM $MgCl_2$, 25 mM HEPES, pH 7.3 and an osmolality of 340 mOsm.). The collected eggs were washed three times with fresh egg buffer. Then the eggs were separated using 30% final concentration of sucrose by centrifuging at 1200 rpm for 20 min. The separated top layers of eggs were collected in a new tube and washed 2–3 times with egg buffer. Then, the eggs were treated with chitinase (1 U/ ml)) for 40 min to digest the egg shell. The chitinase reaction was stopped using L15 medium. After chitinase treatment the embryos were passed through 25 G needle for dissociating into single cells. The dissociated cell suspension was filtered through 5 µm Durapore filter (Millipore). Then the single cell solution was centrifuged for 3 min at 3200 rpm and the pellet was resuspended in a lower volume of L15 medium and seeded on to #1.5 glass bottom petri dish coated with peanut lectin and incubated at 25 C. The cell culture medium was changed every day and at day three post plating we proceeded with the immunocytochemistry.

## Immunocytochemistry protocol

Cells were washed 2x gently with media, followed by a PBS++ (Ca2+ and Mg2+) wash. Then the cells fixed cells with 4% (v/v) PFA in PBS++ (Ca2+ and Mg2+) for 10–20 min at RT. Then cells were washed 3x with PBS. Cells were permeabilized using 0.01% Triton in PBS for 20 min at RT. Then the cells were washed 3x with PBS, and then blocked in 2% BSA for 20–30 min at RT. The primary antibody (anti-green fluorescent protein, REF A11122, Invitrogen by Thermo Fisher Scientific) was directly added the blocking buffer at 1:1000 dilution and incubated at 4 C overnight. The next day the cells were washed 3x with PBS, 10 min each. Then, the secondary antibody (Abberior STAR 488, Cat #2-0012-006-5, Abberior) was used at a dilution of 1:1000 and incubate for 2 hr at RT. The cells were washed 5x with PBS, 10 min each. Then the samples were mounted in Mowiol and stored at 4° C until imaging.

## Imaging and data analysis

### Fluorescent imaging

L4 or young adult worms were anesthetized with 10 mM tetramisole or 5 mM Levamisole on a 5% agarose pad and all live imaging was performed within 60 min. All live fluorescent imaging was performed on a Nikon Eclipse-Ti microscope and with a Plan Apo VC, 60×, 1.40 NA oil or a Plan Apo VC 100 × N.A. 1.40 oil objectives (Nikon). The microscope is equipped with a motorized stage (ASI; PZ-2000), a Perfect Focus System (Nikon), ILas system (Roper Scientific France/PICT-IBiSA, Curie Institute) and uses MetaMorph 7.8.0.0 software (Molecular Devices) to control the camera and all motorized parts. Confocal excitation and detection is achieved using a 100 mW Vortran Stradus 405 nm, 100 mW Cobolt Calypso 491 nm and 100 mW Cobolt Jive 561 nm lasers and a Yokogawa spinning disk confocal scanning unit (CSU-X1-A1N-E; Roper Scientific) equipped with a triple-band dichroic mirror (z405/488/568trans-pc; Chroma) and a filter wheel (CSUX1-FW-06P-01; Roper Scientific) containing BFP (ET-DAPI (49000), GFP (ET-GFP (49002)) and mCherry (ET-mCherry(49008)) emission filters (all Chroma). Confocal images were acquired with a QuantEM:512 SCEMCCD camera (Photometrics) at a final magnification of 110 nm (60x objective), 67 nm (100x objective) per pixel, including the additional 2.0x magnification introduced by an additional lens mounted between scanning unit and camera (Edmund Optics). All EBP-2::GFP imaging was performed at one frame per second (fps) and when necessary Z stacks were taken with 1 µm step size and maximum intensity projections and further processing was performed using ImageJ software (NIH). For Photoactivation of PA-GFP-TBA-1, the red mKate2 fill was used to find the neuron and select the ROI. Photoactivation was performed with the 405 nm laser set at 8–12% laser power and imaged one fps for at least 200 s on the Nikon spinning disk setup. The SRRF optimized image of UNC-44::GFP; MAPH-1, an acquisition of 150 frames at one fps was taken and processed with the nanoJ plugin (*Gustafsson et al., 2016*) in ImageJ.

The PVD neuron morphology, RAB-3::mCherry and MAPH-1 localization were imaged on a LSM 880 confocal Laser Scanning Microscope with Plan Apo 63×/1.2 Imm Corr objective (Zeiss). All other images were acquired using LSM700 (Zeiss) with a 63x/1.40 Oil or 40x/1.30 Oil DIC objective using 488 nm and 555 nm laser lines.

Animal locomotion was imaged on a ZEISS Axio Zoom.V16.

### Analysis of FRAP experiments

To analyze the recovery of fluorescence, 20 pixel round selections were made in regions of bleached and non-bleached GFP fluorescence as well as in the background. The average fluorescence intensity was measured using ImageJ for each frame, the background intensity was subtracted from the bleached and non-bleached areas and normalized to the five frame average prior to the bleach. The bleached area was then divided by the non-bleached area, to correct for bleaching due to imaging and the first frame after bleach was set to 0. To calculate the immobile fraction was calculated as 1-the average recover from 50 to 100 s.

To correct for drift time lapse acquisitions were corrected with the ImageJ plugins *Template Matching* or *StackReg*. Images and time-lapse acquisitions were rotated and if necessary flipped to have the anterior to the left and ventral side down.

### Quantification of RAB-3::mCherry intensity

Z-stacks were acquired with stepsize of 1 µm and sum-projections were made of 6 planes for the axon, the cell body or the proximal anterior dendrite. Selections of 120 × 5 µm (axon), 10 × 10 µm (cell body) and 25 × 2 µm (dendrite) were made in ImageJ and the integrated density was measured.

### TIRF-SIM setup and optical path

TIRF-SIM (structured illumination microscopy) is based on exciting the sample with evanescent sinusoidal illumination patterns and detecting the fluorescence response on a camera. Our custom-built TIRF-SIM setup is shown in *Figure 1—figure supplement 1*. The optical design is based on the use of a digital micromirror device (DMD, DLP V-9501 VIS, ViALUX, pixel pitch 10.8 µm) as spatial light modulator. We employed binary stripe patterns in three orientations and with three phases each. The demagnification of the DMD plane to the sample plane amounts to a factor of 300, resulting in

sinusoidal interference patterns on the sample with a period of approximately 170 nm. This corresponds to an excitation numerical aperture (NA) of 1.44 of the first orders. The DMD also serves as master clock for the trigger protocol. The detailed optical path of the setup comprises the excitation laser with 500 mW power at 473 nm wavelength (gem473, LaserQuantum), followed by a 10x telescope (AC080-010-AML and AC254-100-A-ML, Thorlabs) to expand and collimate the laser beam. A half wave plate (AHWP10M-600, Thorlabs) establishes linear polarization after the DMD. The diffracted light from the DMD is collected by a lens (AC508-500-A-ML, Thorlabs) and passes through a quarter wave plate (AQWP05M-600, Thorlabs), generating circular polarization. Neutral density filters (NDC-50C-2, Thorlabs) provide fine control over the intensities of the individual diffracted beams. A custom aluminum mask is placed in the Fourier plane of the DMD and serves as spatial filter to block unwanted diffraction orders originating from the binary stripe pattern. Only the required first orders pass through and create the sinusoidal intensity pattern on the sample. A segmented polarizer (colorPOL VIS087 BC3 CW01, CODIXX) then generates azimuthal polarization, which is necessary to achieve maximum modulation depth of the sinusoidal pattern at the sample. The combination of the fast-switching DMD with the segmented polarizer implies only mechanically fixed and stable components with no moving parts. The beams are relayed by two lenses (#49–367, Edmund Optics, AC254-300-A-ML, Thorlabs) and then reflected by two identical, but rotated dichroic beam splitters (ZT405/473/561rpc, AHF) to eliminate detrimental polarization effects for s- and p-polarized reflected light. The diffraction spots are projected close to the edge of the back focal plane of the objective (UAPON 100XOTIRF, Olympus), which is mounted onto a z-piezo stage (N-725.2A PIFOC, Physik Instrumente). The beams interfere at the focal plane at a NA of 1.44 and generate the desired sinusoidal illumination pattern on the sample. The emitted fluorescence is collected by the same objective, passes through the dichroic mirror, the tube lens and a combination of emission filters (ZET405/473/561NF, AHF; FF01-503/LP-25, Semrock; HQ515/30 m, Chroma) to reject unwanted excitation light and achieve lowest residual background. A sCMOS camera (Zyla- 4.2 P-CL10, Andor) detects the fluorescence signals in the mode 'rolling shutter global clear external triggering (nonoverlap mode)'. The size of the FOV on the sample is defined by a circular mask in the displayed DMD pattern and amounts to approximately 25 μm.

Each super-resolved image was reconstructed from nine raw widefield images by adapted code of fairSIM in Java as described in *Müller et al. (2016)*. Lines profiles from >11 different randomly chosen cells of at least 3 μm in length (see *Figure 4E*) from reconstructed images were extracted and the dominant frequency was extracted by Fourier decomposition in IgorPro and plotted in *Figure 4E*. Autocorrelation plots were calculated from line profiles across axons in IgorPro (Wavemetrics, Lake Oswego, Oregon).

## Immunoprecipitation experiments

### *C. elegans* liquid culture

Liquid cultures were started with semi-synchronized L1 animals obtained by starvation. Grow 12 worms on 60 mm nematode growth media (NGM) plates with OP50 bacteria until there are no bacteria were left on the plates and the plates were covered with starved L1 animals. All animals were washed off the plates with M9 and transferred to a two liter Erlenmeyer flask containing 500 ml S-medium supplemented with penicillin-streptomycin (5000 U/ml, Life Technologies 15070–63) diluted 1:100, and nystatin suspension (10,000 U/ml, Sigma N1638) diluted 1:1000. A pellet of OP50 *E. coli* bacteria obtained from a 0.5 l overnight culture in lysogeny broth (LB) was added as food source. Animals were incubated at 20°C shaking at 200 rpm for 6 days. To harvest the animals, the culture was transferred to 50 ml conical tubes and cooled on ice for 20 min. Animals were then pelleted by centrifugation (all centrifugation steps in this protocol were performed at 400 g for 2 min at 4°C). After aspirating the supernatant, animals were pooled in a single 50 ml tube, and washed twice in ice-cold M9. After the second wash step, animals were resuspended in 20 ml of ice-cold M9, followed by the addition of 20 ml of ice-cold 60% sucrose in H2O. After vigorous mixing of the sucrose/worm mixture, 4 ml of ice-cold M9 was gently layered on top, and the worms were centrifuged at 400 g for 2 min at 4°C. A layer of animals was now visible on top of the sucrose, while contaminants sedimented at the bottom. The sucrose float steps were performed as quickly as possible, as otherwise the layer of animals failed to form properly. To maximize recovery, 30 ml of supernatant was aspirated from the sucrose float, and distributed into four 50 ml tubes that were

subsequently filled by addition of room-temperature M9. The room temperature M9 allowed the animals to digest OP50 bacteria in their intestine. The four tubes were placed on ice to cool down for 30 min, after which the animals were washed twice in lysis buffer (150 mM NaCl, 25 mM Tris pH 7.5, 1% Triton). During the first wash the animals were again pooled into one tube. After a final wash, *C. elegans* was divided in four 15 ml TPX hard plastic tubes and as much lysis buffer as possible was removed, and the *C. elegans* pellet were frozen in liquid nitrogen and stored at −80˚C.

### *C. elegans* lysis

The *C. elegans* pellets were lysed by sonication using a Bioruptor (diagenode) fitted with a 15 ml tube holder. For this 1 ml of worm pellet was mixed with 1 ml of 2x lysis buffer (50 mM Tris-HCl, pH 7.5, 300 mM NaCl, 2% Triton, 0.5 tablet of protease inhibitor (Roche, 05892791001)). Pellets were gently swirled until thawed. Tubes containing the lysates were placed in the Bioruptor in a bucket filled with ice-water. Samples were lysed 15 times for 30 s, with 30 s intervals. After the third and sixth lysis, lysates were mixed by gently swirling the tube. To remove cellular debris, lysates were distributed to 2 ml Eppendorf tubes, centrifuged at 13,000 rpm for 30 min at 4˚C. The supernatant was incubated with GFP-trap magnetic beads (ChromoTek) for 2 hr at 4˚C. The beads were then washed three times with the lysis buffer and frozen in liquid nitrogen and stored at −80˚C for mass spectrometry analysis.

### Mass Spectrometry

Beads were resuspended in 20 μL of Laemmli Sample buffer (Biorad) and supernatants were loaded on a 4–12% gradient Criterion XT Bis-Tris precast gel (Biorad). The gel was fixed with 40% methanol/10% acetic acid and then stained for 1 hr using colloidal coomassie dye G-250 (Gel Code Blue Stain Reagent, Thermo Scientific). After in-gel digestion, samples were resuspended in 10% formic acid (FA)/5% DMSO and analyzed with an Agilent 1290 Infinity (Agilent Technologies, CA) LC, operating in reverse-phase (C18) mode, coupled to an Orbitrap Q-Exactive mass spectrometer (Thermo Fisher Scientific, Bremen, Germany). Peptides were loaded onto a trap column (Reprosil C18, 3 μm, 2 cm ×100 μm; Dr. Maisch) with solvent A (0.1% formic acid in water) at a maximum pressure of 800 bar and chromatographically separated over the analytical column (Zorbax SB-C18, 1.8 μm, 40 cm ×50 μm; Agilent) using 90 min linear gradient from 7–30% solvent B (0.1% formic acid in acetonitrile) at a flow rate of 150 nl/min. The mass spectrometer was used in a data-dependent mode, which automatically switched between MS and MS/MS. After a survey scan from 350 to 1500 m/z the 10 most abundant peptides were subjected to HCD fragmentation. MS spectra were acquired in high-resolution mode (R > 30,000), whereas MS2 was in high-sensitivity mode (R > 15,000). Raw files were processed using Proteome Discoverer 1.4 (version 1.4.0.288, Thermo Scientific, Bremen, Germany). The database search was performed using Mascot (version 2.4.1, Matrix Science, UK) against a UniProt database (taxonomy *C. elegans*). Carbamidomethylation of cysteines was set as a fixed modification and oxidation of methionine was set as a variable modification. Trypsin was specified as enzyme and up to two miss cleavages were allowed. Data filtering was performed using percolator, resulting in 1% false discovery rate (FDR). Additional filters were search engine rank one and mascot ion score >20.

### Biotin-streptavidin pull-down and western blot

Hek293T cells (authenticated and tested negative for mycoplasma) were co-transfected with BirA together with the indicated bio-mCherry and gfp constructs and incubated for 24–36 hr. Cells were harvested in ice-cold PBS and lysed with lysis buffer (100 mM Tris-HCl pH 7.5, 150 mM NaCl, 1% Triton X-100 and 1x protease inhibitor cocktail). Cell lysates were centrifuged at 13,000 rpm for 10 min and the supernatants were incubated with Dynabeads M-280 (Invitrogen) which were pre-blocked in buffer containing 20 mM Tris, pH 7.5, 20% glycerol, 150 mM NaCl, and 10 μg chicken egg albumin for 30 min followed by two washes of 2 min with wash buffer containing 20 mM Tris, pH 7.5, 150 mM NaCl, and 0.1% Triton X-100. After incubating for 1 hr at 4˚C, beads were washed five times with washing buffer. Samples were eluted with SDS/DTT sample buffer and boiled for subsequent western blot assay. For western blot, samples were loaded onto 10% SDS-PAGE gels and transferred to nitrocellulose membrane. Membranes were blocked with 2% BSA (bovine serum albumin) in PBS/0.05% Tween-20. Primary antibodies (anti-GFP (ab290, abcam) and anti-mCherry (632543,

Clontech)) were diluted in blocking buffer and incubated with the membranes overnight at 4°C, washed three times with PBS/0.05% Tween 20 and incubated with secondary IRDye 680LT or IRDye 800LT antibodies for 45 min at room temperature. Membranes were then washed three times with PBS/0.05% Tween 20 and scanned on Odyssey Infrared Imaging system (LI-COR Biosciences).

For all quantifications, n = the number of individual animals analyzed. Statistical analyses were performed using GraphPad Prism software version 8.0. All bar diagrams represent mean SD statistics was performed using a Kruskal-Wallis test with Dunn's correction, except for the analysis of *Figure 4* and *Figure 1—figure supplement 1* where an unpaired t-test par used.

Data in *Figure 4E* was tested using a non-parametric test with a Nemenyi post-hoc adjustment for multiple comparisons in IgorPro, using a significance level of a = 0.05. The null hypothesis of an equality of the medians was rejected as shown in the pentagram in the inset and set one if the test statistics $q > q_c$ and 0 if otherwise; $q_c$ was determined to be equal to 2.8 for a = 0.05 level of significance and k = 5 groups.

# Acknowledgements

We thanks Mike Boxem and Sander van den Heuvel (Utrecht University, The Netherlands) for advice, *C. elegans* reagents and infrastructure, Yujie Cao for technical assistance with the pull-down experiments, Thijs Koorman, Dipti Rai and Anna Akhmanova for experiments related to this study that were not included, G Ou for sharing unpublished results and Lukas Kapitein, Mithila Burute and Amelie Fréal for discussions and critically reading the manuscript. We acknowledge Gert Jansen, Cori Bargmann, Kang Shen and Kunihiro Matsumo for kind sharing of *C. elegans* strains and reagents. Some strains were provided by the CGC, which is funded by the NIH Office of Research Infrastructure Programs (P40 OD010440) and some by the National Biorescource Project. We thank WormBase for curating and making available data related to *C. elegans*.

This work was funded by the Nederlandse Organisatie voor Wetenschappelijk Onderzoek (NWO) (NWO-ALW-VENI 863.12.001 to MH, NWO-ALW-VICI 865.10.010 to CCH), by the European Research Council (ERC Consolidator Grant 617050 to CCH, ERC starting grant 715243 to MK), HFSP-CDA (CDA00023/2018) to MK, European Union's Horizon 2020 research and innovation programme under the Marie Skłodowska-Curie grant agreement No. 754510 (PROBIST) and by the Chinese Scholarship Council scholarship (CSC) to LH. MK acknowledges financial support from the Spanish Ministry of Economy and Competitiveness (SEV-2015–0522, RYC-2015–17935), from Fundació Privada Cellex, and from Generalitat de Catalunya through the CERCA program and (AGAUR) 2017 SGR 1012.

# Additional information

## Funding

| Funder | Grant reference number | Author |
|---|---|---|
| Chinese Scholarship Counsel | | Liu He |
| Nederlandse Organisatie voor Wetenschappelijk Onderzoek | 863.12.00 | Martin Harterink |
| Nederlandse Organisatie voor Wetenschappelijk Onderzoek | 865.10.010 | Casper C Hoogenraad |
| Ministry of Economy and Competitiveness | SEV-20150522 | Michael Krieg |
| Ministry of Economy and Competitiveness | RYC-2015-17935 | Michael Krieg |
| Fundació Privada Cellex | | Michael Krieg |
| Generalitat de Catalunya | AGAUR 2017 SGR 101 | Michael Krieg |
| European Research Council | 617050 | Casper C Hoogenraad |
| European Research Council | 715243 | Michael Krieg |

| | | |
|---|---|---|
| Human Frontier Science Program | CDA00023/2018 | Michael Krieg |
| H2020 Marie Skłodowska-Curie Actions | 754510 | Ravi Das |

The funders had no role in study design, data collection and interpretation, or the decision to submit the work for publication.

### Author contributions

Liu He, Formal analysis, Validation, Investigation, Visualization, Methodology, Writing - original draft, Project administration; Robbelien Kooistra, Data curation, Formal analysis, Investigation, Visualization, Writing - original draft; Ravi Das, Data curation, Investigation, Visualization; Ellen Oudejans, Data curation, Investigation, Visualization, Methodology; Eric van Leen, Data curation, Formal analysis, Investigation, Methodology; Johannes Ziegler, Supervision, Investigation, Methodology; Sybren Portegies, Investigation, Writing - original draft; Bart de Haan, Investigation, Methodology; Anna van Regteren Altena, Riccardo Stucchi, Data curation, Methodology; AF Maarten Altelaar, Supervision, Performed experiments; Stefan Wieser, Supervision, Methodology; Michael Krieg, Supervision, Writing - review and editing; Casper C Hoogenraad, Supervision, Writing - original draft; Martin Harterink, Conceptualization, Data curation, Formal analysis, Supervision, Funding acquisition, Validation, Investigation, Visualization, Methodology, Writing - original draft, Project administration, Writing - review and editing

### Author ORCIDs

Robbelien Kooistra https://orcid.org/0000-0003-4576-8964
Martin Harterink https://orcid.org/0000-0002-8256-6651

### Decision letter and Author response

Decision letter https://doi.org/10.7554/eLife.55111.sa1
Author response https://doi.org/10.7554/eLife.55111.sa2

## Additional files

### Supplementary files

- Supplementary file 1. Mass spectroscopy identification of co-immunoprecipitated proteins for the unc-119::gfp knock-in immunoprecipitation. Wildtype (N2) and a strain expressing GFP in neurons (OH441) were used as control strains. Find all identified proteins in separate tabs and the analysis of the data in the combined tab. To analyze Affinity Purification Mass Spectrometry Data, a Fold Change score (FC-Score) is calculated based on computing the ratio of average normalized spectral counts (PSMs) in bait purifications versus negative controls using CRAPome (*Mellacheruvu et al., 2013*).

- Supplementary file 2. Overview of all constructs and oligos used in this study.

- Transparent reporting form

### Data availability

All data generated or analysed during this study are included in the manuscript and supporting files.

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
