## [Decision Letter]

**Acceptance summary:**

Using *C. elegans* as a model organism, this study demonstrates that microtubules need to be anchored at the cell cortex of axons and dentrites to prevent kinesin-1 driven microtubule sliding that would otherwise perturb microtubule polarity in neurons. Moreover, the authors have worked out the molecular details of this cortex attachment, providing new interesting insight into the molecular mechanism of the establishment of neuron polarity.

**Decision letter after peer review:**

Thank you for submitting your article "Cortical anchoring of the microtubule cytoskeleton is essential for neuron polarity and functioning" for consideration by *eLife*. Your article has been reviewed by Vivek Malhotra as the Senior Editor, a Reviewing Editor, and three reviewers. The following individuals involved in review of your submission have agreed to reveal their identity: Peter W Baas (Reviewer #1); Vann Bennett (Reviewer #3).

The reviewers have discussed the reviews with one another and the Reviewing Editor has drafted this decision to help you prepare a revised submission.

Summary:

This manuscript describes very interesting work that demonstrates that kinesin-1 generates substantial forces on the microtubules in the axons and dendrites of *C. elegans*. This would cause them to slide if not for membrane-associated attachments of the microtubule bundles. The authors work out in meticulous detail the molecular components of the attachments. They demonstrate when the relevant molecular links are disrupted, the sliding forces of kinesin-1 cause massive disorder of microtubules, which results in morphological defects with functional consequences. The experiments overall are well designed and the results are striking. The observations bring novel insights into the architecture of the microtubule cytoskeleton in neurons. The main areas for improvement concern the presentation. More detail should be added explaining how experiments were done so that the work is more accessible to the non-specialist reader and some claims require modifications to accurately reflect the data as presented.

Essential revisions:

The presentation of the work requires editorial changes. The authors should please describe experiments in sufficient detail, so that they can be reproduced and are also accessible to the non-specialist, should pay attention to the correct use of terms, and make sure that claims are fully supported by their data.

1) It would be helpful for readers unfamiliar with the various polypeptides used here to include their domain maps.

2) Particularly the experiments in Figure 2 are difficult to follow. Please detail how proteins were expressed and isolated, provide conditions for reassembly, and include relevant references. Also, please label relevant polypeptides on the gel panels. The goal should be that someone else can reproduce these experiments. Please ensure that it is always clear whether experiments are performed under conditions of overexpression or whether native protein levels are investigated.

3) Please always state clearly at which stage the animals are when used in experiments (e.g. Figure 3).

4) The wording about the periodicity of the membrane attachments shown in Figure 4 should be tempered a bit. The effect doesn't seem as dramatic as actin rings, for example. Moreover, finding the same periodicity as β-spectrin does not demonstrate that the two are connected ("the complex integrates the microtubule cytoskeleton with the cortical actin/spectrin network"). For this conclusion to be drawn, colocalisation between β-spectrin and the components of the complex should be demonstrated. Please also explain how the neurite was chosen for analysis and in which part of the neurite the periodicity is analysed? Is the neurite fragment close to the cell body, where is the AIS located?

5) With regard to Figure 6 and related wording, the authors should avoid using the term dynamics if they mean sliding. That will confuse some readers.

6) Figure 6: The authors do not discuss that the photoactivated microtubules are visibly weaker in the mutant strains than in the wild type animals. Is there an effect on microtubule dynamics in those mutants? Given that CRMP-2 has been reported to have microtubule assembly-promoting properties (Fukata et al., 2002), the potential impact UNC-44-119-33 complex disruption on microtubule dynamics should be discussed.

7) Figure 7: Does the rescue of microtubule sliding rescue the physiological phenotypes of the UNC-44-119-33 complex mutants, for instance neuronal architecture and worm movements?

---

## [Author Response]

Summary:This manuscript describes very interesting work that demonstrates that kinesin-1 generates substantial forces on the microtubules in the axons and dendrites of *C. elegans*. This would cause them to slide if not for membrane-associated attachments of the microtubule bundles. The authors work out in meticulous detail the molecular components of the attachments. They demonstrate when the relevant molecular links are disrupted, the sliding forces of kinesin-1 cause massive disorder of microtubules, which results in morphological defects with functional consequences. The experiments overall are well designed and the results are striking. The observations bring novel insights into the architecture of the microtubule cytoskeleton in neurons. The main areas for improvement concern the presentation. More detail should be added explaining how experiments were done so that the work is more accessible to the non-specialist reader and some claims require modifications to accurately reflect the data as presented.Essential revisions:The presentation of the work requires editorial changes. The authors should please describe experiments in sufficient detail, so that they can be reproduced and are also accessible to the non-specialist, should pay attention to the correct use of terms, and make sure that claims are fully supported by their data.

As discussed below we have now better described the experiments and modified the wording for some of the conclusions to better reflect the data. Moreover, we changed the color use in Figure 3 and Figure 5 to be consistent to the model and added the model to the figures to help the reader through the data. In addition, in Figure 4, Figure 6 and Figure 7 we added the mammalian names next to the *C. elegans* names.

1) It would be helpful for readers unfamiliar with the various polypeptides used here to include their domain maps.

We added the proteins domain maps to Figure 2.

2) Particularly the experiments in Figure 2 are difficult to follow. Please detail how proteins were expressed and isolated, provide conditions for reassembly, and include relevant references. Also, please label relevant polypeptides on the gel panels. The goal should be that someone else can reproduce these experiments. Please ensure that it is always clear whether experiments are performed under conditions of overexpression or whether native protein levels are investigated.

We clarified the pull-down experiments in the legend and better labeled the relevant proteins and immunoblot conditions in the figure.

3) Please always state clearly at which stage the animals are when used in experiments (e.g. Figure 3).

We indicated the age of the animals in each figure.

4) The wording about the periodicity of the membrane attachments shown in Figure 4 should be tempered a bit. The effect doesn't seem as dramatic as actin rings, for example. Moreover, finding the same periodicity as β-spectrin does not demonstrate that the two are connected ("the complex integrates the microtubule cytoskeleton with the cortical actin/spectrin network"). For this conclusion to be drawn, colocalisation between β-spectrin and the components of the complex should be demonstrated.

We agree with the comment and have toned down the language to reflect the uncertainty about the physical interaction from these superresolved images.

Please also explain how the neurite was chosen for analysis and in which part of the neurite the periodicity is analysed? Is the neurite fragment close to the cell body, where is the AIS located?

*C. elegans* does not have an axon initial segment and thus precludes any analysis accordingly. For all conditions tested, we choose ROIs at random locations along the neurite with a minimum length of 2 μm. However, the labeling structure was variable along the neurite, thus only sections containing a visible periodicity were analyzed for the quantification. We clarified this in the figure legend.

5) With regard to Figure 6 and related wording, the authors should avoid using the term dynamics if they mean sliding. That will confuse some readers.

We changed “dynamics” to “mobility”.

6) Figure 6: The authors do not discuss that the photoactivated microtubules are visibly weaker in the mutant strains than in the wild type animals. Is there an effect on microtubule dynamics in those mutants? Given that CRMP-2 has been reported to have microtubule assembly-promoting properties (Fukata et al., 2002), the potential impact UNC-44-119-33 complex disruption on microtubule dynamics should be discussed.

Indeed, the PA-GFP-tubulin signal was generally weaker in the mutants, which may suggest that there are fewer microtubules. We commented on this in the manuscript. Moreover, we quantified the microtubule dynamics in the *unc-33, unc-119* and *unc-44* mutants and observed small changes compared to wild type (Figure 6—supplement 1).

7) Figure 7: Does the rescue of microtubule sliding rescue the physiological phenotypes of the UNC-44-119-33 complex mutants, for instance neuronal architecture and worm movements?

As the full loss of function mutant for kinesin-1 is not viable, we here depleted kinesin-1 in a cell specific manner (the skin) using lox-sites approach. Therefore, we cannot assess the rescue of the worm locomotion. However we did analyze the microtubule polarity in the *kinesin-1[floxed];unc-33* double mutant, which is not be rescued. We added this to the manuscript (Figure 6—supplement 1D).